# Edge-aware Image Smoothing with Relative Wavelet Domain Representation

**Huiqing Qi**
School of Electronic & Electrical Engineering
Nanyang Technological University
huiqing.qi@ntu.edu.sg

**Xiaoliu Luo** *
College of Science
Chongqing University of Technology
luoxiaoliu@cqut.edu.cn

**Tingting Li**
School of Mathematics & Information Science
Zhengzhou University of Light Industry
52205500006@stu.ecnu.edu.cn

**Fang Li** *
School of Mathematical Sciences,
Key Laboratory of MEA(Ministry of Education)
& Shanghai Key Laboratory of PMMP
East China Normal University
fli@math.ecnu.edu.cn

## Abstract

Image smoothing is a fundamental technique in image processing, designed to eliminate perturbations and textures while preserving dominant structures. It plays a pivotal role in numerous high-level computer vision tasks. More recently, both traditional and deep learning-based smoothing methods have been developed. However, existing algorithms frequently encounter issues such as gradient reversals and halo artifacts. Furthermore, the smoothing strength of deep learning-based models, once trained, cannot be adjusted for adapting different complexity levels of textures. These limitations stem from the inability of previous approaches to achieve an optimal balance between smoothing intensity and edge preservation. Consequently, image smoothing while maintaining edge integrity remains a significant challenge. To address these challenges, we propose a novel edge-aware smoothing model that leverages a relative wavelet domain representation. Specifically, by employing wavelet transformation, we introduce a new measure, termed **R**elative **W**avelet **D**omain **R**epresentation (**RWDR**), which effectively distinguishes between textures and structures. Additionally, we present an innovative edge-aware scale map that is incorporated into the adaptive bilateral filter, facilitating mutual guidance in the smoothing process. This paper provides complete theoretical derivations for solving the proposed non-convex optimization model. Extensive experiments substantiate that our method has a competitive superiority with previous algorithms in edge-preserving and artifact removal. Visual and numerical comparisons further validate the effectiveness and efficiency of our approach in several applications of image smoothing.

## 1 Introduction

Edge-preserving smoothing (EPS) represents a fundamental challenge in computer vision Luo et al. (2024b; 2025; 2024a) and computational graphics. The objective of image smoothing is to eliminate minor perturbations and non-essential textures while maintaining prominent edges and structural features Qi et al. (2024a). EPS has garnered significant attention due to its wide range of applications, including image decomposition Song et al. (2017); Liu et al. (2019), image texture removal Xu et al. (2012), high dynamic range (HDR) tone mapping Anand Swamy & Shylashree (2023), detail enhancement Xu et al. (2022b), texture transfer Gupta et al. (2022), and clip-art compression artifacts removal Li et al. (2024). Over the past few decades, numerous image smoothing algorithms have been developed, which can be broadly classified into local information-based methods, global information-based methods, and deep learning-based approaches. However, existing methods often face limitations related to edge preservation and the reduction of visual artifacts such as halos He et al. (2012) and gradient reversal Gastal & Oliveira (2012), as illustrated in Figure 1. To address

---

*Corresponding Author

these challenges, Liu et al. (2018) has explored embedding bilateral filters within least squares models. More recently, the incorporation of soft clustering into bilateral filtering has been introduced Yang et al. (2021). Additionally, studies Li et al. (2023); He et al. (2022) have combined relative total variation with bilateral filters.

Drawing inspiration from multi-model embedding methods, we propose a novel mutual guidance EPS model. Specifically, we first derive a guidance image by solving a non-convex function, which leverages a newly introduced relative wavelet domain representation. We then integrate the proposed edge-aware scale map of the guidance image into a bilateral filter. This edge-awareness technique enhances the algorithm's ability to preserve edges effectively. In a nutshell, the primary contributions of this work are as follows: **(1)** We introduce a novel global information-based relative measure with wavelet transformation, termed **R**elative **W**avelet **D**omain **R**epresentation (**RWDR**), which more effectively distinguishes textures from primary structures and preserves weaker edges. **(2)** We integrate the proposed edge-aware scale map into an adaptive bilateral filter for mutual guidance in smoothing, which significantly reduces gradient reversal and halo artifacts. **(3)** Extensive experimental results demonstrate that our model outperforms competing algorithms in both visual and numerical evaluations across a range of smoothing applications. This indicates that our proposed model achieves an improved balance between smoothing strength and edge preservation.

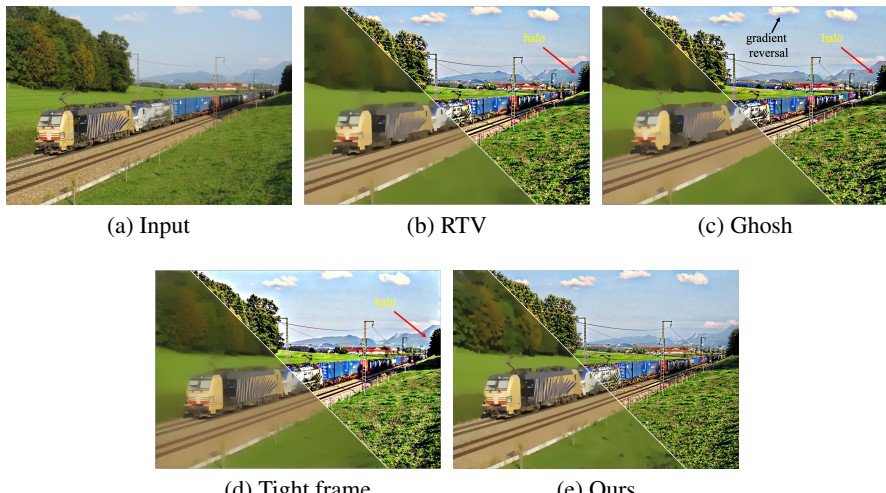

(a) Input        (b) RTV        (c) Ghosh

(d) Tight frame        (e) Ours

Figure 1: Image detail enhancement. The left-bottom part of each plot presents the smoothed image, and the right-top part shows the detail enhancement image that is boosted by four detail layers. The detail layer is extracted by subtracting the smoothed image from the input image. Results of (b), (c), and (d) suffer halos and grandient reversal artifacts.

All theoretical derivations and more EPS application experiments that have been omitted for space appear in the supplementary material.

## 2 RELATED WORK

**Local information-based models.** Methods of this category take the weighted average of neighboring pixels' values in a local window as an output. These approaches are known as filter-based methods, which include bilateral filter (BF) Tomasi & Manduchi (1998), joint bilateral filter Petschnigg et al. (2004), and rolling guidance filter Zhang et al. (2014a). There are also other filter-based methods such as weighted median filter Ma et al. (2013); Zhang et al. (2014b), guided filterHe et al. (2012), tree filter Bao et al. (2013), recursive filter Tu & Chien (2021), and mutually guided filtering Guo et al. (2017). The fixed size of a local window at the central pixel limits the performance of local information-based methods. Therefore, several scale-adaptive bilateral filters Xu et al. (2022a); Ruhela et al. (2022); Ghosh et al. (2019); Song et al. (2019) and structure/scale-aware filtering methods Kaplan & Erer; Gupta et al. (2022) have been proposed to improve the performance. However, they cause halo artifacts He et al. (2012) and gradient reversals Gastal & Oliveira (2012) in EPS application tasks.

**Global information-based models.** Algorithm in this category solve the image smoothing problem via finding a global solution to the specific objective function. The well-known approaches are the total variation based methods, which include weighted least squares smoothing Min et al. (2014), gradient $L_1$ norm smoothing Pang et al. (2015), and gradient $L_0$ norm smoothing Xu et al. (2011). Many other global information-based approaches have recently been proposed, including the relative total variation (RTV) smoothing Xu et al. (2012) and its variants Liu et al. (2016); Yu et al. (2022); Li et al. (2023); Qi et al. (2022). Methods built on RTV are sensitive to edges aligned with horizontal and vertical directions. Directional relative total variation (DRTV) smoothing Zhou et al. (2019) has been proposed to overcome this drawback. Generally, global information-based approaches are based on solving an extensive linear system to the global optimization objective function, which is high computational and time cost. Global information-based methods have a tradeoff between time cost and smoothing performance.

**Deep learning-based models.** These methods are built on different deep convolutional neural networks (DCNN) for smoothing Zhu et al. (2019). These methods are end-to-end smoothing frameworks Lu et al. (2018); Pan et al. (2018). The main drawback of the end-to-end based methods is that obtaining the paired training data is challenging. Therefore, unsupervised learning smoothing Fan et al. (2018) has been proposed to overcome the lack of paired training data. Some other deep learning methods for smoothing include filter operator approximation methods Chen et al. (2017); Li et al. (2019) via deep neural networks. Generally, methods in this category also suffer from artifacts.

## 3 BACKGROUND

### 3.1 ADAPTIVE BILATERAL FILTER

The adaptive bilateral filter (ABF) proposed by Ghosh et al. (2019) builds upon the traditional bilateral filter (BF) Tomasi & Manduchi (1998) for image smoothing. Unlike the BF, which utilizes a fixed spatial kernel, the ABF employs a box function where the spatial kernel's scale varies at each pixel. Ghosh et al. (2019) emphasize that the local scale of the range kernel plays a crucial role in preserving texture details. As a result, they introduced a scale map applied to each pixel in the filter. Given an input image $I$, the output image $S$ in ABF is expressed as:

$$S_p = \frac{\sum_{q \in R_p(p)} G_{\sigma_r}(||I_p - I_q||)I_q}{\sum_{q \in R_p(p)} G_{\sigma_r}(||I_p - I_q||)}, \tag{1}$$

where $G_{\sigma_r}$ denotes a Gaussian range kernel with a standard deviation $\sigma_r$. Stated simply, the aggregation at pixel $p = (x, y)$ is perfored over a local window $R_p(p)$ centered at $p$, and

$$R_p(p) = (x + k_1, y + k_2), -\omega_p \le k_1, k_2 \le \omega_p, \tag{2}$$

where $\omega_p$ denotes the scale at pixel $p$, which is from the scale map.

The ABF has a fast approximation version that can achieve $\mathcal{O}(1)$ complexity in Ghosh et al. (2019). However, it produces results with gradient reversals and halos in the image detail enhancement. The gradient reversals are caused by the sharpened edges in the smoothed output, while halos are typically caused by large $\sigma_1$ and $\sigma_2$ values, it can be observed in Figure 1(c).

### 3.2 WAVELET TRANSFORM TIGHT FRAME

The tight frame is an important concept in wavelet transformation, widely used in image processing. For more details, refer to Shen (2010); Dong et al. (2010). Firstly, we present the definition of a tight frame. $\mathcal{H}$ be a Hilbert space. A sequence $x_n \subset \mathcal{H}$ can be considered a tight frame for $\mathcal{H}$ if

$$||x||^2 = \sum_n | < x, x_n > |^2, \quad for \; any \; x \in \mathcal{H}. \tag{3}$$

There are two associated operators, which are the analysis operator and the synthesis operator. And they are expressed as

$$W : x \in \mathcal{H} \to \{\langle x, x_n \rangle\} \in \ell_2(\mathbb{N}), \quad W^{\mathrm{T}} : a_n \in \ell_2(\mathbb{N}) \to \sum_n a_n x_n \in \mathcal{H}. \tag{4}$$

Then, $x_n$ is a tight frame if and only if $W^{\mathrm{T}}W = I$, where $I : \mathcal{H} \to \mathcal{H}$ is the identical operator. $\mathbb{N}$ denotes the natural number set. The discrete tight frame is widely used based on the theory of tight frames. Cai et al. (2014) used an undecimal tight frame in the image restoration. Given a filter

$a \in \ell_2(\mathbb{Z})$, the linear convolution operator $C_a : \ell_2(\mathbb{Z}) \to \ell_2(\mathbb{Z})$ as

$$[C_a v](n) := [a * v](n) = \sum_{k \in \mathbb{Z}} a(n-k)v(k), \ \forall v \in \ell_2(\mathbb{Z}). \tag{5}$$

Given a set of filters $\{a_i\}_{i=1}^m \subset \ell_2(\mathbb{Z})$, the corresponding analysis operator and synthesis operator are written as follows

$$W = [C_{a_1(-.)}^{\mathrm{T}}, C_{a_2(-.)}^{\mathrm{T}}, \cdots, C_{a_m(-.)}^{\mathrm{T}}]^{\mathrm{T}}, \quad W^{\mathrm{T}} = [C_{a_1}, C_{a_2}, \cdots, C_{a_m}]. \tag{6}$$

The rows of $W$ forms a tight frame for $\ell_2(\mathbb{Z})$ if and only if $W^{\mathrm{T}}W = I$. $\mathbb{Z}$ is the rational number set. When constructing tight frames, the unitary extension principle proposed a full constraint condition Ron & Shen (1997). One of typically used full unitary extension principle conditions Ron & Shen (1997) is expressed as

$$\sum_{i=1}^m \sum_{n \in \mathcal{Z}^2} a_i(k+n)a_i(n) = \delta_k, \quad for \ any \ k \in \mathcal{Z}^2. \tag{7}$$

With such full unitary extension principle condition, the well-known B-spline tight frame filters are used in many image applications Chai & Shen (2007); Cai et al. (2008; 2009). These filters are written as follows

$$a_1 = \frac{1}{4}(1,2,1)^{\mathrm{T}}, \ a_2 = \frac{\sqrt{2}}{4}(1,0,-1)^{\mathrm{T}}, \ a_3 = \frac{1}{4}(-1,2,-1)^{\mathrm{T}}. \tag{8}$$

The listed filters satisfy the full condition Equation 7. For the image smoothing task, we only need the 2D framelet filters $\{a_i \bigotimes a_j\}_{i,j=1}^m$ that is generated via the tensor product $\bigotimes$ of filters in Equation 8. The tight frame is sensitive to the weak edges that locate along diagonal directions. However, the tight frame also causes halos in the output, as shown in Figure 1(d). In summary, the tight frame is a specific class in wavelet transformation.

## 4 METHODOLOGY

### 4.1 RELATIVE WAVELET DOMAIN REPRESENTATION

Given an image $I$, the non-convex smoothing problem with relative wavelet representation is expressed as:

$$\arg\min_S \sum_p (S_p - I_p)^2 + \lambda \frac{D(p)}{W(p) + \varepsilon}, \tag{9}$$

where $\lambda$ denotes a weight value that controls the smoothing strength. $\varepsilon$ is a positive small constant, which avoids division by zero. $\frac{D(p)}{W(p)+\varepsilon}$ is the proposed **relative wavelet domain representation**. $D(p)$ and $W(p)$ are written as:

$$D(p) = \sum_{q \in R(p)} g_{p,q} \sum_{k=1}^K |(w_k S)_q|, \quad W(p) = \sum_{k=1}^K \left| \sum_{q \in R(p)} g_{p,q}(w_k S)_q \right|, \tag{10}$$

$$g_{p,q} \propto \exp\left(-\frac{(x_p - x_q)^2 + (y_p - y_q)^2}{2\sigma^2}\right), \tag{11}$$

where $q$ belongs to $R(p)$ that is a rectangle region centred at pixel $p$. $(x_*, y_*)$ denote the pixel index. $w_k$ denotes tight frame filters of wavelet transformtation in Equation 8. $K$ denotes the number of tight frame filters. The smoothing performance of the proposed model is highly dependent on the RWDR, which effectively distinguishes between main structures and textures. The RWDR serves as a regularization term, assigning larger values to texture regions and smaller values to the structural areas.

We compare $D(p)/(W(p) + \varepsilon)$ with other regularization term maps in Figure 2. The left-up region of each image denotes the corresponding regularization term map, while the right-bottom region of each image presents the corresponding final texture removal result. Obviously, Figure 2 shows RoG Cai et al. (2017), imRTV Yu et al. (2022), and RTV methods tend to divide some textures into edges. The dRTV blurred these junction regions between textures and structures and also loses some main structures. In contrast, the map of our RWDR clearly identifies the structures and the textures. Meanwhile, comparing these final smoothed results in Figure 2, our approach provides a better edge-preserving smoothed image than other methods.

Based on Equation 9 - Equation 11, we obtain the intermediate smoothed image via the following optimization objective function, expressed as

$$\min_S \sum_p (S_p - I_p)^2 + \lambda \frac{\sum_{q \in R(p)} g_{p,q} \sum_{k=1}^K |(w_k S)_q|}{\sum_{k=1}^K |\sum_{q \in R(p)} g_{p,q}(w_k S)_q| + \varepsilon}. \tag{12}$$

The soluion to Equation 12 can be obtained by solving a linear system with the preconditioned conjugate gradient (PCG) via iteraive mode. More details are in **Appednix A**. The global solution to Equation 12 is the inermediate smoothed result $S^t$, used in the following mutually guided adaptive bilateral filter.

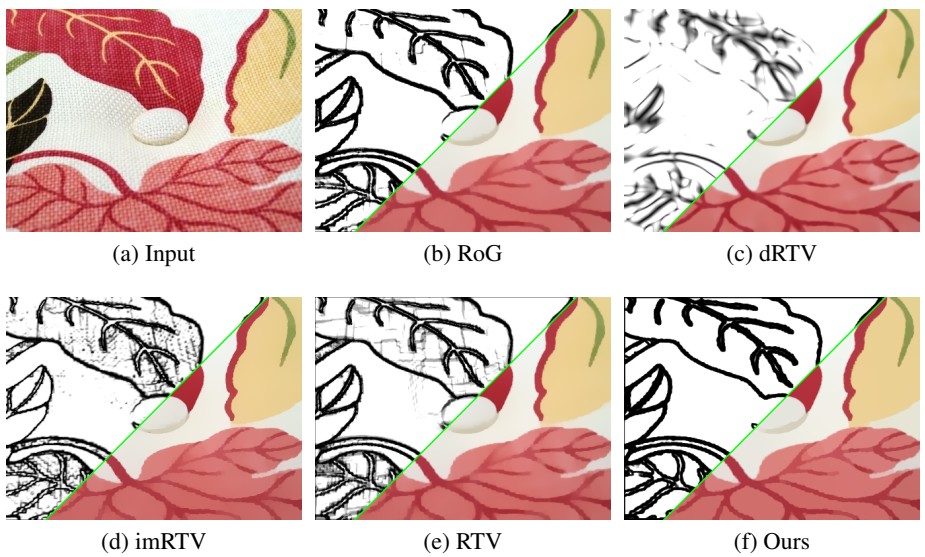

(a) Input            (b) RoG            (c) dRTV

(d) imRTV         (e) RTV         (f) Ours

Figure 2: Relative wavelet domain representation comparison. Results of (b) RoG, (c) dRTV, (d) imRTV, (e) RTV, (f) Ours. The left-up part of each image shows the relative feature map, and the right-bottom part is the corresponding smoothed image.

### 4.2 MUTUALLY GUIDED ADAPTIVE BILATERAL FILTER

Drawing inspiration from muGIF Guo et al. (2017), we use the mechanism of mutual guidance between the proposed RWDR and adaptive bilateral filter. It means that the outputs of RWDR and adaptive bilateral filters act as cross inputs simultaneously. Therefore, the proposed mutually guided adaptive bilateral filter is built on Equation 1, expressed as:

$$F_p^{t+1} = \frac{\sum_{q \in N_p(p)} G_{\sigma_r}(||S_p^t - S_q^t||)S_q^t}{\sum_{q \in N_p(p)} G_{\sigma_r}(||S_p^t - S_q^t||)}, \tag{13}$$

where $t$ denotes the iterative number. $N_p(p)$ is a local window that centered at $p = (i, j)$. We also use Equation 2 to obtain the scale map, expressed as

$$N_p(p) = \{(i + k_1, j + k_2) : -\omega_p \le k_1, k_2 \le \omega_p\}. \tag{14}$$

It is important to note the subtle distinction between $N_p(p)$ and $R_p(p)$ in Equation 2. The term $R_p(p)$ is dynamic and evolves with each iteration. The subsequent scale map is dependent on the previously smoothed image; thus, if the previous iteration produces a poorly smoothed image, the subsequent iteration is likely to exacerbate the error, often resulting in the blurring of weak edges. However, we propose a novel edge-aware approach to derive the scale map, effectively mitigating the issue of blurred weak edges and enhancing edge-preserving capabilities. Additionally, the Fourier approximation of the mutually guided adaptive bilateral filter achieves $\mathcal{O}(1)$ computational complexity, as demonstrated in **Appendix B.**

### 4.3 EDGE-AWARE SCALE MAP

The scale map plays a critical role in Equation 14. Ghosh et al. (2019) introduce a method for deriving the scale map through the erosion of the gradient map. In contrast, we propose a novel

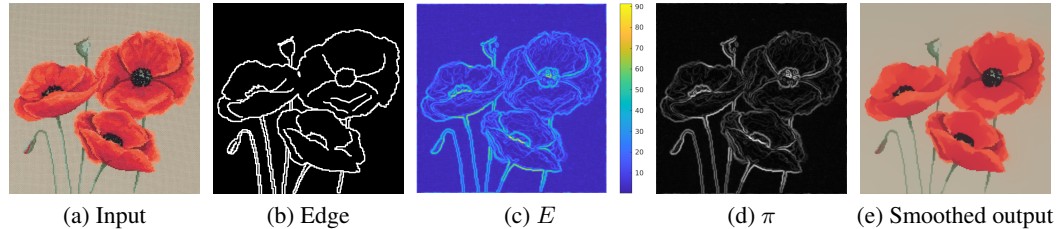

(a) Input      (b) Edge      (c) $E$      (d) $\pi$      (e) Smoothed output

Figure 3: Illustration of an edge-aware map. (b) Edges, (c) $E$ map. Notice that $\pi$ map (d) covers all edges in (b). $\pi$ acts as an edge-aware map that can improve edge-preserving ability, as shown in (e).

edge-aware method for computing the scale map. Given an input image $I$, the intermediate feature map $E$ is defined as:

$$E = \sum_{i \in N_8(p)} |G_{\sigma_e} \nabla_i I|. \tag{15}$$

$G_{\sigma_e}$ denotes a Gaussian range kernel. $N_8(p)$ is a 8-neighborhood set that centered at pixle $p$. $\nabla_i$ is the gradient operator. The $E$ map facilitates the extraction of key structural information, as illustrated in Figure 3(c). Subsequently, we define the general gradient magnitude as follows:

$$g_h = \sqrt{(\nabla_i I)^2 + (\nabla_j I)^2}, \quad i = \{1,2,3,4\}, \; j = 9 - i. \tag{16}$$

$g_h$ can capture the local edge feature information. Then, a coefficient based on $g_h$ can be obtained by

$$m = \frac{1}{4} \sum_{h=1}^{4} \frac{1}{g_h \exp(g_h/\gamma^2) + \varepsilon}. \tag{17}$$

$\gamma$ is a constant. $\varepsilon$ is a positive value, which avoids division by zero. $m$ provides a small value in the structure regions while providing a large value in the texture ones. $m$ acts as a weight map to obtain the edge-aware feature map $\pi$, which is expressed as

$$\pi = E(1 - m). \tag{18}$$

We show $\pi$ map in Figure 3(d). As can be observed in Figure 3(b), it is ground truth edges. Obiviously, $\pi$ provides small values in texture regions while large values are in structure regions. We use such a $\pi$ map to obtain the scale map, which can help to improve the edge-preserving ability in the smoothed output. The scale $\omega_p$ map should be designed to satisfy that it is small along sharp edges and large in smooth regions. Then, $\omega_p$ is expressed as

$$\omega_p = r_{min} + (r_{max} - r_{min}) \exp(-\lambda_e \pi_p^2). \tag{19}$$

$\lambda_e$ is a positive value to control the transition rate from $r_{min}$ to $r_{max}$. We constrain $\omega_p$ in an interval $[r_{min}, r_{max}]$. $\omega_p$ is rounded off to be the final edge-aware scale map. The effects of scale maps obtained by using Equation 19 and Ghosh model are shown in Figure 4.

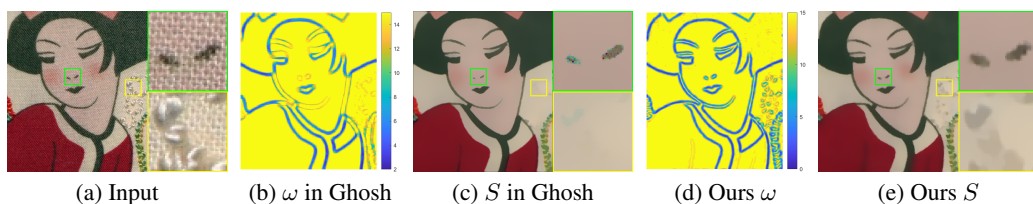

(a) Input     (b) $\omega$ in Ghosh     (c) $S$ in Ghosh     (d) Ours $\omega$     (e) Ours $S$

Figure 4: Adaptive scale maps ($r_{max} = 15$). (a) The input image, (b) $\omega$ in Ghosh model, (c) smoothed image corresponding (b), (d) scale map $\omega$, (e) smoothed image of our model. Note that $\omega$ is able to maintain more edge information than that of $\omega$ in (b). The more edge information in $\omega$, the more edges and details can be preserved. Refer to the enlarged areas of (c) and (e).

## 4.4 ALGORITHM

For the sake of completeness, we briefly present our final model, which is built upon the RWDR and the mutually guided edge-aware adaptive bilateral filter. The proposed method is formulated as

follows:

$$
\begin{cases}
S^t & = \arg\min_{S^t} \sum_p (S_p^t - F_p^{t-1})^2 + \lambda \frac{\sum_{q \in R(p)} g_{p,q} \sum_{k=1}^K |(w_k S^t)_q|}{\sum_{k=1}^K |\sum_{q \in R(p)} g_{p,q}(w_k S^t)_q| + \varepsilon}, \\
F_p^{t+1} & = \frac{\sum_{q \in N_p(p)} G_{\sigma_r}(||S_p^t - S_q^t||) S_q^t}{\sum_{q \in N_p(p)} G_{\sigma_r}(||S_p^t - S_q^t||)}.
\end{cases}
\tag{20}
$$

When $t$ is assigned at 1, $F^0$ is initialized to the input image $I$. $F^t$ and $S^t$ update each other via mutually guided. We draft the scheme of the mutually guided edge-aware RWDR smoothing model in Algorithm 1.

---

**Algorithm 1** Mutually Guided Edge-aware RWDR

---

**Input:** Image $I$, $\sigma_e$, $\gamma$, $\lambda_e$, $r_{min}$, $r_{max}$, $\sigma$, $\varepsilon$, $\lambda$, $K$, $a_i$, $\sigma_r$.
**Initialization:** $F^0 = I$.
 1: Calculate $E$ via Equation 15;
 2: Calculate $m$ via Equation 17;
 3: Calculate $\pi$ with $m$ and $E$ via Equation 18;
 4: Calculate $\omega_p$ with $\pi$ in Equation 19;
 5: **for** $t = 1$ **to** $T$ **do**
 6:    Update $S^t$ with $F^{t-1}$ via Equation 12;
 7:    Update $F^{t+1}$ with $S^t$ and $\omega_p$ via Equation 13;
 8: **end for**
**Output:** Smoothed image $S$.

---

## 5 EXPERIMENTS

To assess the performance and effectiveness of our approach, we conduct a comparative analysis against several existing methods, including BF Tomasi & Manduchi (1998), L0 Xu et al. (2011), WLS Min et al. (2014), RGF Zhang et al. (2014a), ILS Liu et al. (2020), EP/SP Liu et al. (2021), dRTV Jeon et al. (2016), DRTV Zhou et al. (2019), imRTV Yu et al. (2022), RoG Cai et al. (2017), muGIF Guo et al. (2017), RTV Xu et al. (2012), Ghosh Ghosh et al. (2019), Resnet Lim et al. (2017), and VDCNN Kim et al. (2016). We utilize five no-reference image quality objective evaluation metrics: BRISQUE Mittal et al. (2012), PIQE Venkatanath et al. (2015), SSEQ Liu et al. (2014), ILNIQUE Zhang et al. (2015), and CEIQ Yan et al. (2019) to compare performance on image detail enhancement and HDR tone mapping tasks. The smaller values of BRISQUE and PIQE denote higher-quality images. While the larger values of SEQ, ILNIQUE, and CEIQ indicate higher-quality images. Meanwhile, CEIQ is specifically designed to measure the effects of the HDR tone mapping task. We utilized the original code and recommended parameters provided by the respective authors for comparison. For all the evaluated approaches, the parameters were fine-tuned to achieve an optimal balance between smoothing performance and edge preservation. Our code will be available on my github.

**Texture removal.** We evaluate the texture removal performance of our model in comparison to other methods, as illustrated in Figure 5. Notably, the enlarged regions in (b), (c), (d), (f), (l) of Figures 5 reveal instances of incomplete texture removal. Additionally, the enlarged yellow rectangles in Figures 5(e) and (h) indicate that the detailed edges are blurred. Furthermore, Figures 5(g), (j), (k), and (m) demonstrate suboptimal performance in edge preservation. Although Figures 5(i) and (n) exhibit competitive edge-preserving capabilities, Figure 5(n) introduces halo artifacts along the edges, as indicated by the enlarged magenta region. In contrast, our proposed method achieves a superior balance between texture smoothing and edge preservation compared to other approaches.

**Image detail enhancement.** It aims to enhance high-frequency regions by incorporating a detail layer into the input image. The core of this technology involves extracting the high-frequency detail layer by subtracting the smoothed image from the original input. Consequently, the effectiveness of detail enhancement is closely linked to the quality of the smoothed image, particularly with respect to prominent edges. Two extreme scenarios can arise: one where blurred edges introduce halo artifacts in the detail-enhanced image, and another where sharp edges result in gradient reversals in the enhanced output. In this study, we enhance the details by adding four detail layers to the

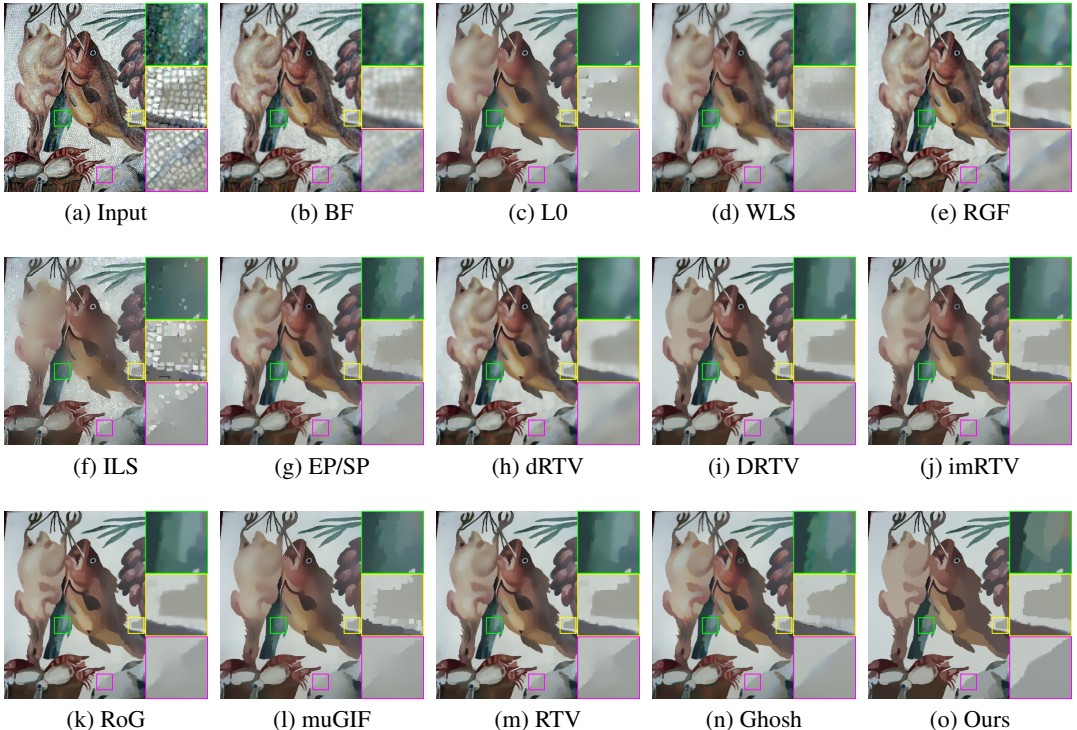

Figure 5: (a) The input image. Its textures are removed by (b) BF, (c) L0, (d) WLS, (e) RGF, (f) ILS, (g) EP/SP, (h) dRTV, (i) DRTV, (j) imRTV, (k) RoG, (l) muGIF, (m) RTV, (n) Ghosh, (o) Ours. Refer to these highlighted boxes, our method can get a better tradeoff between the smoothing strength and the edge-preserving than other algorithms.

original image. We present the smoothed images alongside their corresponding detail-enhanced versions in Figure 6. In the enlarged magenta and green regions of each image, Figures 6(b) - (f) exhibit halo artifacts along the edges. Figure 6(g) displays a slight halo effect within the green enlarged rectangle. Notably, our model demonstrates a significant reduction in the occurrence of halo artifacts. Additionally, as indicated by the enlarged rectangle in the lower-left corner of each image, our model effectively preserves weak edges. We present corresponding numerical results in Table 1. One can see that our mode obtains the best index values.

Table 1: No-reference image quality evaluation metrics of Figure 6.

| Methods | BRISQUE↓ | PIQE ↓ | SSEQ ↑ | ILNIQUE↑ |
|---------|----------|--------|--------|----------|
| L0 | 10.6974 | 35.6176 | 24.7566 | 117.81 |
| ILS | 23.5255 | 37.0118 | 17.7950 | 122.43 |
| DRTV | 8.5623 | 33.5745 | 25.1130 | 124.46 |
| RoG | 14.9437 | 34.6882 | 24.5738 | 124.32 |
| muGIF | 20.6328 | 36.6683 | 25.1297 | 121.44 |
| dRTV | 15.4301 | 25.1630 | 9.9543 | 124.49 |
| Ours | **7.6912** | **24.9118** | **26.7940** | **128.84** |

**Compression artifact removal.** The technology for removing compression artifacts is employed to mitigate blocking artifacts in compressed clip-art images. When an image is compressed at a low bit rate using standard JPEG encoding, it may result in compression artifacts along sharp edges and staircase artifacts in homogeneous regions. In this study, we employ a $10\%$ compression rate for the clip-art images. Figure 7 illustrates the smoothed results obtained from various methods. Notably, the upper right part of Figure 7(a) displays the original clip-art image, while the lower left part shows the compressed version. Observations indicate that Figures 7(b), (c), and (d) exhibit blurred edges

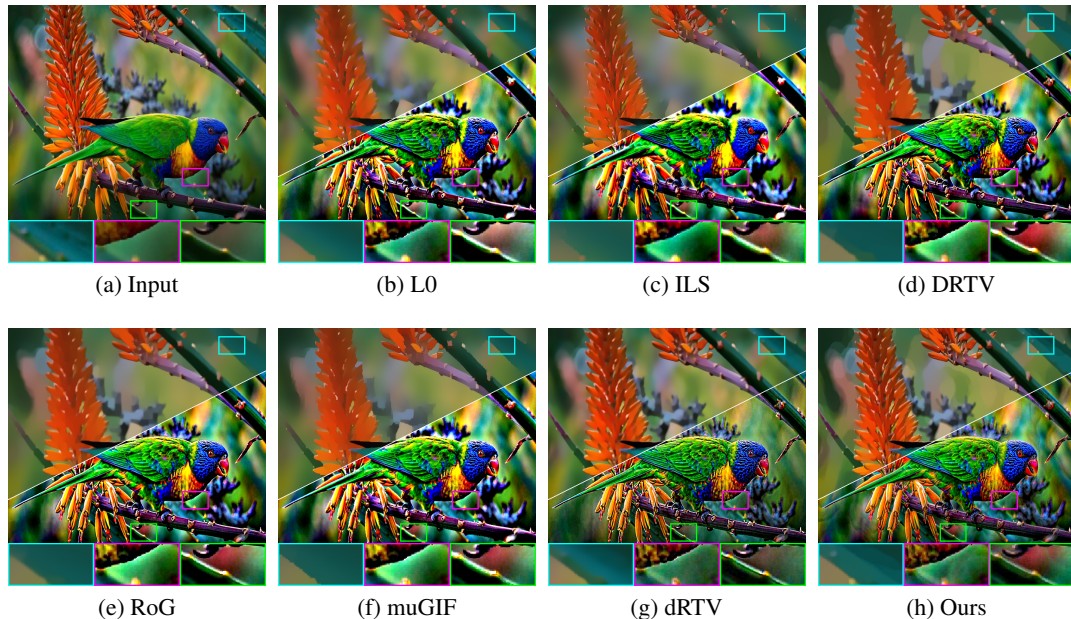

Figure 6: Image detail enhancement. (a) The input image. It is enhanced with the detail layer $4\times$ boosted by (b) L0, (c) ILS, (d) DRTV, (e) RoG, (f) muGIF, (g) dRTV, (h) Ours. The cyan highlighted box denotes the details of the smoothed image, magenta and green boxes are the details of the enhanced image. In these highlighted rectangles, our method can reduce halo artifacts. In addition, it performs better in edge-preserving than other state-of-the-art algorithms.

in the smoothed images, as highlighted in the right box of each image. Additionally, Figures 7(e) and 7(f) present staircase artifacts near the edges. Furthermore, Figure 7(g) displays colorful halos along the edges, as indicated by the left red-highlighted box. In contrast, our approach effectively eliminates staircase artifacts and halos while demonstrating strong edge-preserving capabilities. We also employ two widely recognized quality metrics, PSNR and SSIM, to assess the images resulting from compression artifact removal, as presented in Table 2. Higher values of PSNR and SSIM indicate superior quality in the smoothed images. Our proposed model demonstrates the best performance, achieving the highest PSNR value of 27.10 and an SSIM value of 0.991. In summary, both numerical and visual comparisons indicate that the proposed algorithm outperforms others in the removal of compression artifacts.

Table 2: PSNR(dB) and SSIM comparison.

| Metric | WLS | RoG | RGF | muGIF | EP/SP | imRTV | BF | Ghosh | L0 | ILS | DRTV | Ours |
|---|---|---|---|---|---|---|---|---|---|---|---|---|
| PSNR | 20.55 | 26.43 | 26.21 | 25.97 | 26.68 | 26.80 | 21.44 | 20.51 | 25.91 | 21.67 | 25.80 | **27.10** |
| SSIM | 0.937 | 0.978 | 0.974 | 0.977 | 0.980 | 0.979 | 0.924 | 0.894 | 0.976 | 0.909 | 0.975 | **0.991** |

**Ablation Study.** To assess the capability of RWDR in distinguishing between textures and structures, we conduct an ablation study on RWDR in Figure 8. The model deployed without RWDR has mistreated texture as structure, leading to removing texture uncleanly. In contrast, To assess the capability of the edge-aware scale map in edge preservation, we conduct an ablation study on the edge-aware scale map in Figure 9. The model deployed without the edge-aware scale map has smoothed textures cleanly while making main structures and edges lost and blurred. As observed in the enlarged red- and blue-highlight areas, our model achieves the best visual effects.

## 6 CONCLUSIONS AND LIMITATIONS

This study introduces a mutually guided edge-aware smoothing model based on relative wavelet domain representation. The RWDR serves as a novel measure for effectively differentiating between textures and structures. The proposed edge-aware scale map is integrated into an adaptive bilateral filter, providing mutual guidance to the RWDR during the smoothing process. The solution of the

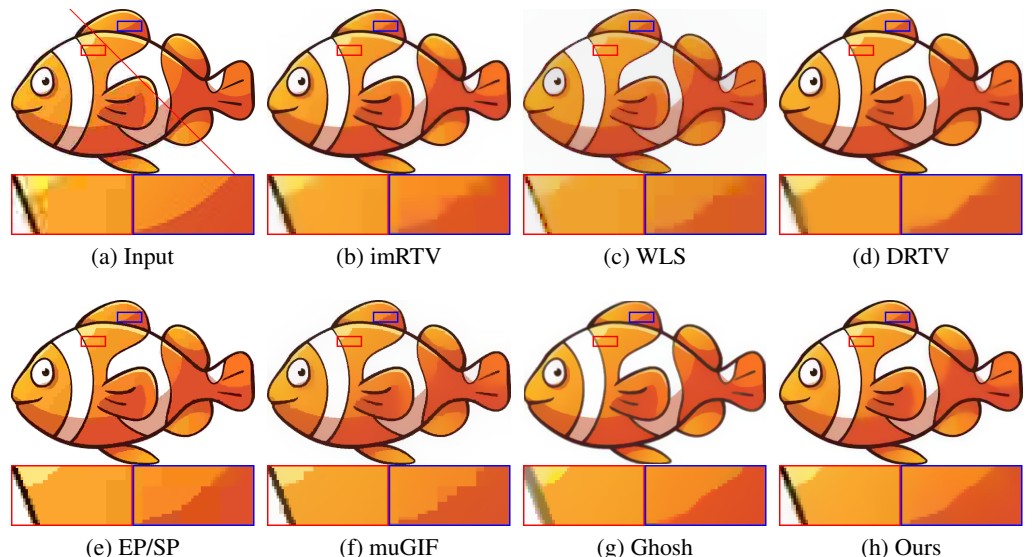

Figure 7: Compresion artifact removal results. The left-bottom part of (a) is the compressed JEPG image, and the right-top part is the original image. Results are obtained by (b) imRTV, (c) WLS, (d) DRTV, (e) EP/SP, (f) muGIF, (g) Ghosh, (h) Ours. Referring to the marked boxes, results of (b), (c), and (d) suffer from blurred edges. Results of (e), (f), and (g) have staircase artifacts.

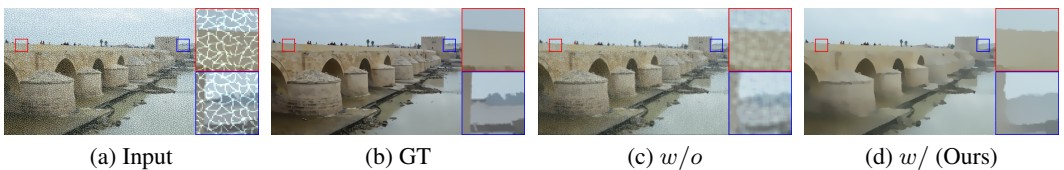

Figure 8: Visual effects of the ablation experiment on the RWDR.

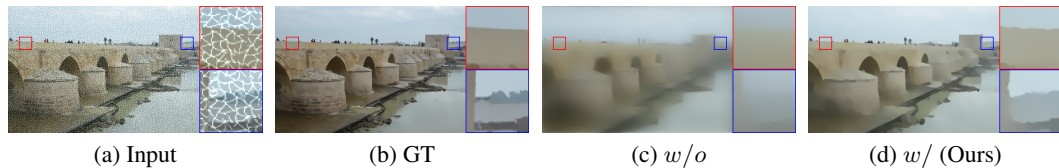

Figure 9: Visual comparison of the ablation experiment on the edge-aware scale map.

proposed model is supported by a complete theoretical guarantee. Extensive comparisons demonstrate that the proposed method outperforms existing algorithms in mitigating gradient reversals, staircase artifacts, and halos. Whether compared to traditional image smoothing techniques or deep learning-based approaches, our method consistently achieves a superior balance between smoothing strength and edge preservation.

**Limitations.** Traditional smoothing methods, including our model, are limited in their ability to address long-range texture dependencies. This implies that our model has constraints when dealing with irregular multiscale textures. A promising direction for future work would be to integrate the proposed algorithm into a neural network, such as **Vision Mamba** Zhu et al. (2024), for enhanced image smoothing. Exploring the potential of combining traditional models with neural networks could provide valuable insights into achieving more effective texture smoothing while maintaining superior edge preservation.

ACKNOWLEDGEMENT

This work is supported by Natural Science Foundation of Shanghai (No. 22ZR1419500), Science and Technology Commission of Shanghai Municipality (No. 22DZ2229014), Fundamental Research Funds for the Central Universities. This paper is also supported by Chongqing University of Technology Research Startup Funding (2023ZDZ032); Science and Technology Research Program of Chongqing Municipal Education Commission (KJQN202401138); Science and Technology Innovation Key R&D Program of Chongqing (CSTB2024TIAD-STX0022).

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

This supplementary document paper provides an in-depth account of our mathematical analysis and presents additional numerical results. The structure of this supplement is organized as follows. Section A.1 presents a mathematical analysis of the solution to the proposed non-convex problem utilizing a relative wavelet domain representation. In Section A.2, we discuss the computational complexity and convergence properties of the Fourier approximation applied to the mutually guided edge-aware adaptive bilateral filter. Section B provides detailed descriptions of the parameter settings, runtime analysis, and presents five additional experimental results as well as fail scenarios.

# A  THEORETICAL ANALYSIS OF RWDR

## A.1  NUMERICAL SOLUTION TO RWDR

This subsection primarily presents the numerical solution for the proposed RWDR non-convex problem. To begin, we revisit the formulation of the RWDR , which is expressed as:

$$\arg\min_{S} \sum_{p}(S_p - I_p)^2 + \lambda \text{RWDR}(p). \tag{21}$$

And we have

$$\text{RWDR}(p) = \frac{D(p)}{W(p) + \varepsilon} = \frac{\sum_{q \in R(p)} g_{p,q} \sum_{k=1}^{K} |(w_k S)_q|}{\sum_{k=1}^{K} |\sum_{q \in R(p)} g_{p,q}(w_k S)_q| + \varepsilon}. \tag{22}$$

Then, we note that when the two sum operators exchange the order, there is no differencen from the original formal. Therefore, we can get

$$\sum_{p} \frac{D(p)}{W(p) + \varepsilon} = \sum_{p} \frac{\sum_{k=1}^{K} \sum_{q \in R(p)} g_{p,q} |(w_k S)_q|}{\sum_{k=1}^{K} |\sum_{q \in R(p)} g_{p,q}(w_k S)_q| + \varepsilon}$$
$$= \sum_{k=1}^{K} \sum_{p} \frac{\sum_{q \in R(p)} g_{p,q} |(w_k S)_q|}{\sum_{k=1}^{K} |\sum_{q \in R(p)} g_{p,q}(w_k S)_q| + \varepsilon}. \tag{23}$$

To simplify, we let

$$R_k = \sum_{p} \frac{\sum_{q \in R(p)} g_{p,q} |(w_k S)_q|}{\sum_{k=1}^{K} |\sum_{q \in R(p)} g_{p,q}(w_k S)_q| + \varepsilon}$$
$$\simeq \sum_{q} \sum_{p \in R(q)} \frac{g_{p,q} |(w_k S)_q|^2}{\sum_{k=1}^{K} |\sum_{q \in R(p)} g_{p,q}(w_k S)_q| + \varepsilon} \frac{1}{|(w_k S)_q| + \varepsilon_s} \tag{24}$$
$$= \sum_{q} u_{kq} z_{kq} (w_k S)_q^2,$$

where $\varepsilon_s$ denotes constant value avoid division by zero. The $u_{kq}, z_{kq}$ can be written as follows:

$$\begin{cases} u_{kq} = \sum_{p \in R(q)} \frac{g_{p,q}}{W(p) + \varepsilon} = (G_\sigma * \frac{1}{\sum_{k=1}^{K} |G_\sigma * (w_k S)| + \varepsilon})_q, \\ z_{kq} = \frac{1}{|(w_k S)_q| + \varepsilon_s}. \end{cases} \tag{25}$$

$G_\sigma$ is a Gaussian filter with a standard deviation $\sigma$. Therefore, we can obtain the matrix form of Equation 21, expressed as:

$$(V_S - V_I)^{\text{T}}(V_S - V_I) + \lambda \sum_{k=1}^{K}(V_S^{\text{T}} C_k^{\text{T}} U_k Z_k C_k V_S). \tag{26}$$

$V_s, V_I$ are the vector form of output image $S$ and input image $I$, respectively. $C_k$ is the Toeplitz matrices from the discrete gradient operators with a forward difference. $U_k$ and $Z_k$ are diagonal matrices with values of $U_k[i, i] = u_{ki}$ and $Z_k[i, i] = z_{ki}$, respectively.

The form of Equation 26 can be solved via a special iterative optimization procedure. It boils down to solving a linear system via iterative mode, which is expressed as

$$(\mathbf{1} + \lambda R^t)V_S^{t+1} = V_I, \tag{27}$$

where $\mathbf{1}$ is the identity matrix. $R^t = \sum_{k=1}^{K}(C_k^{\text{T}} U_k^t Z_k^t C_k)$ and $\mathbf{1} + \lambda R^t$ is the symmetric positive definite Laplacian matrix. In this case, many efficient solvers are available for it. We adopt the preconditioned conjugate gradient (PCG) to solve it.

A.2 FOURIER APPROXIMATION OF MUTUALLY GUIDED ABF

In this section, we present the Fourier approximation of mutually guided adaptive bilateral filter with our edge-aware scale map. For completeness, we review Equation 13, expressed as

$$F_p^{t+1} = \frac{\sum_{q \in N_p(p)} G_{\sigma_r}(||S_p^t - S_q^t||)S_q^t}{\sum_{q \in N_p(p)} G_{\sigma_r}(||S_p^t - S_q^t||)}, \tag{28}$$

where $G_{\sigma_r}$ is Gaussian range kernel with a standard deviation $\sigma_r$. To simplify, let

$$\psi(t) = G_{\sigma_r}(t) = \exp(-\frac{t^2}{2\sigma_r^2}). \tag{29}$$

In this study, we adopt the Fourier expansion of Equation 29 to approximate the adaptive bilateral filter, which deployed with an edge-aware scale map. Specifically, Equation 29 can be approximated in the following format:

$$\hat{\psi}(t) = \sum_{n=-N}^{N} c_n \exp(\tau nvt), \tag{30}$$

where $\tau^2 = -1$, $v = \pi/T$. $N$ is the order of Fourier expansion, and $c_n$ is the coefficient. $t$ denotes the difference of pixel values in the local window $\Omega_p$. We can get the range of $t \in \{-T, \ldots, 0, \ldots, T\}$, obtained via:

$$T = \max_{p \in S^t} \max_{q \in \Omega_p} |S^t(q) - S^t(p)|. \tag{31}$$

Therefore, we take the approximation used in Equation 28, which results in the approximation of $\hat{F}^{t+1}(p)$. It is written as:

$$\hat{F}^{t+1}(p) = \frac{H(p)}{Q(p)}, \tag{32}$$

where

$$H(p) = \sum_{q \in \Omega_p} \hat{\psi}(S^t(q) - S^t(p))S^t(q), \tag{33}$$

and

$$Q(p) = \sum_{q \in \Omega_p} \hat{\psi}(S^t(q) - S^t(p)). \tag{34}$$

Then, we further have

$$\begin{cases} H(p) = \sum_{n=-N}^{N} c_n \exp(\tau nvS^t(p))h_n(p), \\ Q(p) = \sum_{n=-N}^{N} c_n \exp(\tau nvS^t(p))q_n(p), \end{cases} \tag{35}$$

where $h_n(p)$ and $q_n(p)$ are expressed as follows:

$$\begin{cases} h_n(p) = \sum_{q \in \Omega_p} S^t(q)\exp(\tau nvS^t(q)), \\ q_n(p) = \sum_{q \in \Omega_p} \exp(\tau nvS^t(q)). \end{cases} \tag{36}$$

In this study, $h_n(p)$ and $q_n(p)$ can be calculated by adding each pixel value of $\Omega_p$. However, the computation cost is expensive. To simplify, we get $h_n(p)$ and $q_n(p)$ via recursive. Firstly, we let $m(q)$ be the integrated element of pixel $p$. Then, we have

$$m(q) = S^t(q)\exp(\tau nvS^t(q)). \tag{37}$$

And also can obtain the integral image $M(p)$ at the center pixel $p$.

$$M(p) = M(x, y) = \sum_{k_1=1}^{x} \sum_{k_2=1}^{y} m(k_1, k_2), \tag{38}$$

where $(x, y)$ is the coorddinate of the center pixel $p$, $k_1, k_2$ are the coordinates from the edge-aware scale map $\omega_p$.

we compute Equation 38 by using recursion at pixel $(x + 1, y + 1)$ as follows:

$$M(x + 1, y + 1) = m(x + 1, y + 1) + M(x + 1, y) + M(x, y + 1) - M(x, y). \tag{39}$$

By adopting this way, given an edge-aware scale map $\omega_p$, we can get $h_n(p)$ as follows:

$$\begin{aligned} h_n(p) = &M(x + \omega_p, y + \omega_p) - M(x - \omega_p - 1, y + \omega_p) \\ &- M(x + \omega_p, y - \omega_p - 1) + M(x - \omega_p - 1, y - \omega_p - 1). \end{aligned} \tag{40}$$

Similarly, $q_n(p)$ can be calculated. In summary, we can implement Equation 28 at $\mathcal{O}(1)$ computational complexity in a recursive manner. We consider the approximation error that is defined as

$$||F^{t+1} - \hat{F}^{t+1}||_\infty = \max\{|F^{t+1}(p) - \hat{F}^{t+1}(p)| : t \in \{0, \dots, T\}\}. \tag{41}$$

It shows the largest difference in the pixxel values between the original and approximating outputs. Since $H(p)$ and $Q(p)$ are calculated exactly, the error mainly comes from the approximation of the Gaussian range kernel. Therefore, we have

$$||\psi - \hat{\psi}||_\infty = \max\{|\psi(t) - \hat{\psi}(t)| : t \in \{0, \dots, T\}\}. \tag{42}$$

According to previous work Ghosh et al. (2019), it gives the guarantee condition of convergence for Equation 41 bounded by Equation 42. The condition is expressed as

$$||F^{t+1} - \hat{F}^{t+1}||_\infty \leq \frac{2\theta\varepsilon}{\eta - \varepsilon}, \tag{43}$$

where $\eta = \frac{1}{(2r_{max}+1)^2}$ and $\theta$ is the dynamic range and $r_{max}$ is the maximum value in $\omega_p$. In conclusion, we have the complete theoretical guarantee for the convergence of Fourier approximation in our mutually guided edge-aware ABF.

# B  ADDITIONAL NUMERICAL RESULTS

## B.1  PARAMETER SETTINGS

For the scheme presented in Algorithm 1, the input parameters are $\sigma_e$, $T$, $\gamma$, $\lambda_e$, $r_{min}$, $r_{max}$, $\sigma$, $\varepsilon$, $\lambda$, $K$, $a_i$, and $\sigma_r$ that need to be initialized in advance. $\sigma_e$, $\gamma$, $\lambda_e$, $\varepsilon$, and $K$ are fixed parameters. We empirically found that smoothed results are not sensitive to these parameters. The default reasonable fixed settings are $\sigma_e = 3$, $\gamma = 10/255$, $\lambda_e = 4$, $\varepsilon = 5e - 4$, and $K = 4$, which are suitable for each test image. We have set $r_{min} = 0$ to guarantee that the minimized value in the edge-aware scale map is close to 1. However, $r_{max}$ is changed as the width of textures for different images. The recommended value of $r_{max}$ lies in the range $[5, 15]$. We empirically find that different strengths of weak edges can be preserved with different values of $\sigma_r$. The recommened range of $\sigma_r$ is $[10, 30]$. For our experiments, we have used $\sigma = 3$, and $\lambda$ is in the range $[0, 0.008]$. We utilized $a_1$ and $a_3$ of Equation 8 in our all experiments. It is worth noting that it is hard to automatically determine parameters in the image smoothing problem. Therefore, we recommend that users tune parameters in real-world applications via our proposed approach.

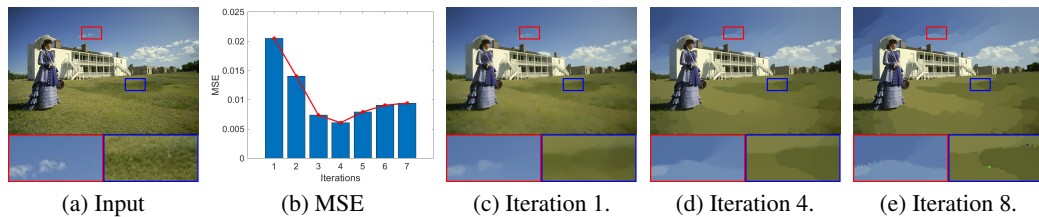

|  (a) Input  |  (b) MSE  |  (c) Iteration 1.  |  (d) Iteration 4.  |  (e) Iteration 8.  |

Figure 10: Iteration number exploration. (a) The input image, (b) the mean square difference of gradients between adjacent outputs, (c) the smoothed image after 1 iteration, (d) the smoothed image after 4 iterations, (e) the smoothed image after 8 iterations. We note that there has a minimum of 4 iterations in (b), which denotes the best output. We also get the same conclusion by (d).

The parameter $T$ is crucial for the quality of smoothed images. A larger value of $T$ can cause over-smoothing, while a small value can not remove all textures. We can obtain a satisfactory result with a suitable stop in the smoothing process. The effects of $T$ on the final output in our model are presented in Figure 10. We have conducted our method with 8 iterations. Figure 10(b) shows the mean square difference of gradient magnitudes between adjacent smoothed results. It is evident that the mean square difference is minimum when $T = 4$. Figure 10(c)-(e) present the smoothed results of our model at $T = 1, 4, 8$. Figure 10(c) shows it cannot ensure sufficient smoothing strength with a small iteration number. However, a large iteration value causes artifacts and outliers, as shown in Figure 10(e). The best-smoothed result with $T = 4$ is shown in Figure 10(d). Considering the effect of over-smoothing, we set the max iteration number $T = 4$ in all our experiments.

## B.2 RUNNING TIME

We compare the running time of different methods under parameters recommended by corresponding-ing authors. We slightly tune the parameters of those methods to achieve satisfactory performance. This paper uses a uniform metric Luo et al. (2021) for smoothing strength to compare running time fairly. Given an input image $I$ and final adjacent smoothed outputs $S^{t-1}, S^t$, the metric is defined as

$$Re = (||S^t - S^{t-1}||_F^2)/||I||_F^2, \tag{44}$$

where $F$ denotes Frobenius norm. We conduct the speed comparison on the image with a size of $768 \times 1024 \times 3$ under $Re = 0.00001$. The average running time is shown in Table 3. Our method is faster than muGIF, RoG, Ghosh. and EP/SP while slower than ILS, RGF, and RTV. Our model needs to solve a large linear system, not achieving the same speed as ILS, RGF, and RTV.

Table 3: Running Time (in Seconds) of Different Methods.

| Methods | ILS | RGF | RTV | muGIF |
|---|---|---|---|---|
| Time (s) | **0.59** | 0.75 | 4.65 | 10.31 |
| Methods | RoG | EP/SP | Ghosh | Ours |
| Time (s) | 11.15 | 14.42 | 7.60 | 6.08 |

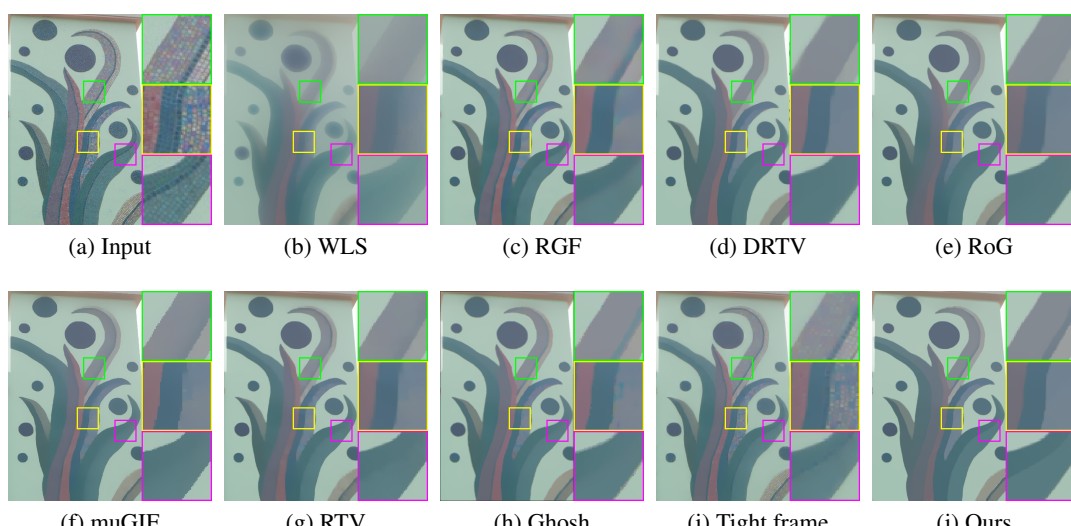

Figure 11: Image texture removal results of different methods. Parameters of all compared methods are tuned to achieve a similar smoothing effect in the highlighted green boxes. Results of (c), (f), (g), (h), and (i) suffer from texture removal not thoroughly, referring to marked yellow boxes. For highlighted magenta rectangles, except for the tight frame and our method, the rest models have blurred edges.

## B.3 TEXTURE REMOVAL

The objective of texture removal is to distinguish and eliminate complex textures while preserving the primary structures. Figure 11 presents the smoothed outcomes produced by various methods. Each plot in Figure 11 contains three enlarged regions, where it is evident that models such as WLS, RGF, DRTV, RoG, muGIF, RV, and Ghosh blur edges or structural details. Although the tight frame model better preserves edges, it fails to completely remove textures. In contrast, the proposed model effectively preserves sharp edges while thoroughly eliminating textures. Further comparisons between our model and a deep learning-based method are shown in Figure 12. VDCNN Kim et al. (2016), a deep-learning-based texture removal method, produces the result shown in Figure 12(c). While VDCNN reduces textures, it does not entirely remove them, as observed in the two enlarged regions. Figure 12(b) displays the result from the RTV model, which successfully removes textures but at the cost of blurring edges, as shown in the two enlarged rectangles. In contrast, our model clearly removes textures while preserving the primary structures. The key advantage of our approach is its ability to achieve a superior balance between smoothing strength and edge preservation compared to other algorithms.

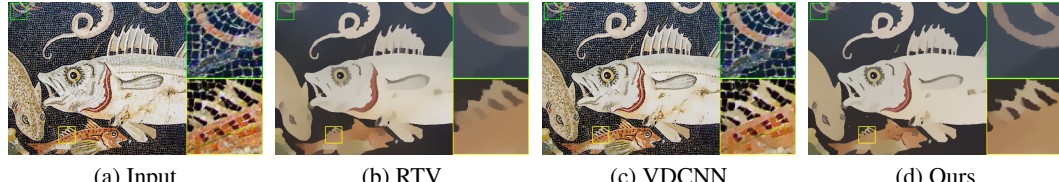

(a) Input        (b) RTV        (c) VDCNN        (d) Ours

Figure 12: (a) The input image. Texture removal results of (b) RTV, (c) VDCNN, (d) Ours. As observed in the highlighted boxes, our model can effectively preserve edges, which substantiates that the proposed method can get a better tradeoff between smoothing strength and edge-preserving than the compared methods.

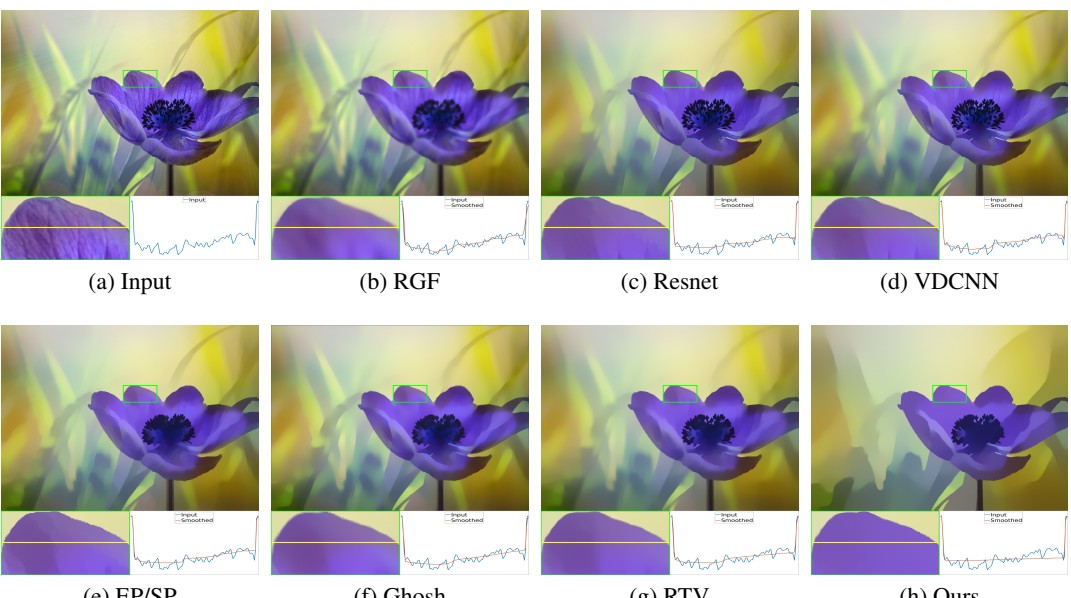

(a) Input      (b) RGF      (c) Resnet      (d) VDCNN

(e) EP/SP      (f) Ghosh      (g) RTV      (h) Ours

Figure 13: Image smoothed results of (b) RGF, (c) Resnet, (d) VDCNN, (e)EP/SP, (f) Ghosh, (g) RTV, (h) Ours. The right-bottom part of each image is the 1D signal smoothed result corresponding to the yellow line in the green marked box. The blue line is the input 1D signal, while the red is the smoothed result. We note that our method has a better-smoothed output than other algorithms. Meanwhile, the proposed model eliminates staircase artifacts.

## B.4 NATURAL IMAGE SMOOTHING

The task of natural image smoothing serves as an effective benchmark for evaluating a model's ability to balance smoothing intensity with edge preservation. Figure 13 displays the results of natural image smoothing along with corresponding 1D signal comparisons. The lower-left rectangle in each image is an enlarged view of the region within the green-marked box, while the lower-right rectangle shows the 1D signal comparison along the yellow line within each enlarged region. In Figure 13(b), the RGF method produces a smoothed output that suffers from excessive blurring. Figures 13(c) to 13(g) exhibit staircase artifacts in homogeneous regions, which are also evident as abrupt changes along the middle of the red lines in the 1D plots. In contrast, our proposed method effectively eliminates staircase artifacts, delivering the most refined smoothing result among all the methods compared.

## B.5 DETAIL ENHACNCEMENT

Figure 14 illustrates additional results of detail enhancement. Figure 14(a) shows the original input image. Notably, Figures 14(b) and 14(d) exhibit gradient reversals in the output images, as highlighted within the enlarged green rectangles. Figures 14(c), 14(e), and 14(f) show similar gradient reversal effects. Additionally, halo artifacts along the edges are present in all the methods shown in Figures 14(b) through 14(f). It is also worth noting that the two deep learning-based methods (ResNet and VDCNN) fail to achieve an optimal balance between edge preservation and smoothing.

In contrast, the proposed method produces results that are comparable to, and in some cases better than, the other approaches in this scenario. We present corresponding numerical results in Table 4. One can see that our mode obtains the best index values.

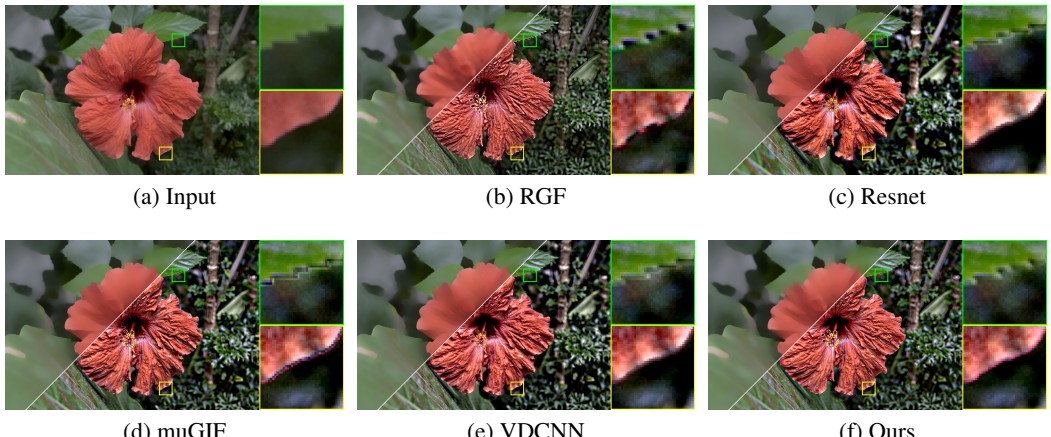

| (a) Input | (b) RGF | (c) Resnet |
| --- | --- | --- |
| (d) muGIF | (e) VDCNN | (f) Ours |

Figure 14: (a) The input image. Image detail enhancement results of (b) RGF, (c) Resnet, (d) muGIF, (e) VDCNN, (f) Ours. We use four detail layers to boost the input image in this case. Note that our model can reduce halo artifacts and weaken the gradient reversals (referring to the marked boxes).

Table 4: No-reference image quality evaluation metrics of Figure 14.

| Methods | BRISQUE↓ | PIQE ↓ | SSEQ ↑ | ILNIQUE↑ |
| --- | --- | --- | --- | --- |
| RGF | 28.8617 | 30.1807 | 23.7323 | 117.56 |
| ResNet | 22.3801 | 36.4881 | 26.4587 | 120.12 |
| muGIF | 16.0524 | 30.2385 | 19.8289 | 123.25 |
| VDCNN | 28.9499 | 35.4522 | 26.1938 | 123.85 |
| Ours | **9.6503** | **26.8804** | **31.5996** | **126.78** |

## B.6 HDR TONE MAPPING

HDR tone mapping is a key application of image smoothing, requiring the extraction of a detail layer Liu et al. (2018) from the HDR input image. The smoothed result is then used to generate a low dynamic range (LDR) image through nonlinear mapping. The final tone-mapped output is produced by combining the nonlinearly transformed smoothed image with the detail layer. However, since HDR tone mapping relies on the high-frequency detail layer, it is also prone to gradient reversals and halo effects. Figure 15 presents the HDR tone mapping results from different methods. In Figure 15(a), gradient reversals and artifacts along the edges are evident, as highlighted by the green and yellow boxes. Figure 15(d) suffers from pronounced halos, especially in the two highlighted regions, where strong white halos are visible along the edges. The other methods avoid gradient reversal artifacts and halo effects in their outputs. We present corresponding numerical results of this HDR tone mapping task in Table 4. One can see that our mode obtains the best index values. Overall, our proposed method demonstrates competitive performance compared to other approaches in this task.

## B.7 TEXTURE TRANSFERRING

The goal of texture transfer is to generate target images with new texture patterns, which can include regular, near-regular, and irregular textures. One of the main challenges in deep learning methods for texture removal is the difficulty of acquiring paired training datasets. Texture transfer offers a solution to this problem. Training datasets can be created by transferring the texture pattern from a source image (as shown in Figure 16(a)) to a target image (as shown in Figure 16(b)). In essence, the texture transfer process, as described in Ghosh et al. (2019), involves three steps. First, both the source and target images are smoothed. Second, the texture layer is extracted by subtracting the smoothed source image from the original source image, followed by extracting the texture pattern from a local window (illustrated in Figure 16(a)). Finally, the texture patch is resized to fit the target image. The transferred output is obtained by adding the smoothed target image to the resized texture

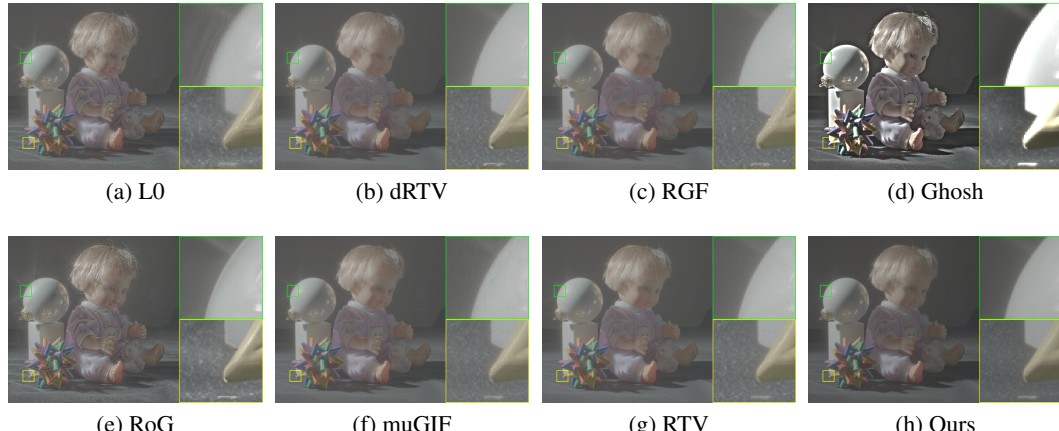

Figure 15: HDR tone mapping. Results of (a) L0, (b) dRTV, (c) RGF, (d) Ghosh, (e) RoG, (f) muGIF, (g) RTV, (h) Ours. We note that the L0 model produces staircase artifacts near edges. The result of (d) shows heavy halo artifacts. On the contrary, our model has comparable results in HDR tone mapping over the other methods.

Table 5: No-reference image quality evaluation metrics of Figure 15.

| Methods | BRISQUE↓ | PIQE ↓ | SSEQ ↑ | ILNIQUE↑ | CEIQ↑ |
|---------|----------|--------|--------|----------|-------|
| L0 | 24.8086 | 28.6853 | 12.3523 | 129.59 | 2.6947 |
| dRTV | 23.7972 | 26.9161 | 16.8464 | 129.16 | 2.6867 |
| RGF | 24.3680 | 27.2629 | 15.6265 | 126.34 | 2.7728 |
| Ghosh | 23.7344 | 38.8357 | 15.3685 | 125.49 | 2.0085 |
| RoG | 28.1463 | 28.8937 | 12.1442 | 135.53 | 2.8496 |
| muGIF | 22.8831 | 32.4163 | 13.2606 | 132.94 | 2.6515 |
| RTV | 20.7134 | 30.0749 | 16.1376 | 133.17 | 2.6804 |
| Ours | **14.8449** | **24.8736** | **17.9714** | **135.69** | **2.8867** |

layer, as shown in Figure 16(c). More recently, Qi et al. (2024b) have created a paired dataset for edge-preserving smoothing tasks via leveraging texture transfer technology.

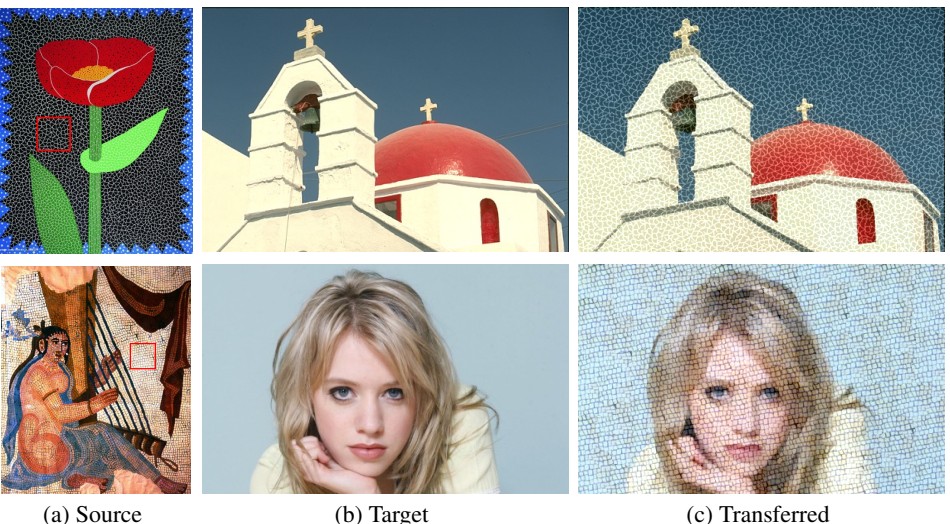

(a) Source          (b) Target          (c) Transferred

Figure 16: Texture transfer. (a) Source images, (b) Target images, (c) Texture transferred images. We copy textures of the marked boxes in (a) to fusion with target images (b), and the final results are shown in (c).

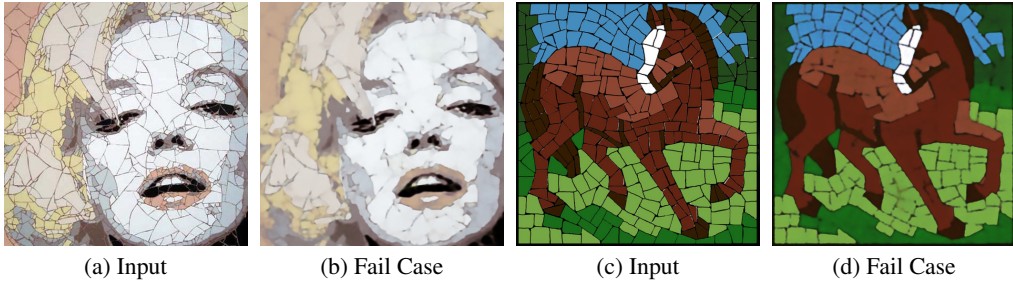

|  (a) Input | (b) Fail Case | (c) Input | (d) Fail Case |

Figure 17: Visual effects of our model on the fail scenario.

## B.8 THE ANALYSIS OF FAILURE CASES

As can be seen from Figure 17, textures in Figure 17(a) and (c) are irregular and relatively large in scale. Therefore, smoothing these textures faces the challenge of dealing with the long-range dependency problem. In other words, it is difficult to remove them from the context of the current pixel. Our approach can remove some of these textures, but cannot completely remove them. Our model's performance relies on the edge-aware scale map, while these long-range textures are mistreated as edges and kept. Even our model is deployed with Wavelet domain features for smoothing. It is still to fail in this scenario.

