# OpenReview forum: "Edge-aware Image Smoothing with Relative Wavelet Domain Representation"
_ICLR.cc/2025/Conference — ICLR 2025 Poster_

### Official Review · Reviewer_M1ZN · 2024-11-02

**Soundness:** 3
**Presentation:** 3
**Contribution:** 2
**Rating:** 6
**Confidence:** 4

**Summary:**

In this article, the author reviews image smoothing methods based on local information, global information, and deep learning, and discusses the limitations of current image smoothing techniques when dealing with image textures and image structural edges. To address this issue, first, the author proposes a novel edge-aware smoothing model that more effectively distinguishes between image textures and image structures through relative wavelet domain representation (RWDR). Second, the author reintroduces edge-aware scale maps into bilateral filters to improve image edges during the smoothing process. Finally, the author demonstrates the superiority of this method in texture preservation and artifact removal after image smoothing through comprehensive theoretical derivations and experimental results compared to other algorithms.

**Strengths:**

1.This paper proposes a relative wavelet domain representation and an edge-aware smoothing model, achieving certain progress in image smoothing technology; 2.The paper utilizes extensive theoretical proofs to establish a mathematical model for the relative wavelet domain. The experimental validation is well-supported by theory; 3.The writing of this paper is relatively fluent and conforms to the standards of English academic writing.

**Weaknesses:**

1.In the experimental part of this paper, there is a predominance of qualitative analysis of images. However, due to the significant subjective factors inherent in qualitative experiments, supplementing with more quantitative experiments would enhance the persuasiveness of the results; 2.Image smoothing operations, as one of the fundamental image processing tasks, play a crucial role in various visual tasks. However, the paper seems to lack exploration of specific visual tasks (for example, in super-resolution tasks, the textures and structures preserved after image smoothing are vital for image reconstruction).

**Questions:**

1.In the experimental part of this paper, there is a predominance of qualitative analysis of images. However, due to the significant subjective factors inherent in qualitative experiments, supplementing with more quantitative experiments would enhance the persuasiveness of the results; 2.Image smoothing operations, as one of the fundamental image processing tasks, play a crucial role in various visual tasks. However, the paper seems to lack exploration of specific visual tasks (for example, in super-resolution tasks, the textures and structures preserved after image smoothing are vital for image reconstruction).

---

> ### Author Response · Authors · 2024-11-22
>
> We greatly appreciate Reviewer M1ZN for their positive assessment and constructive feedback. We address each question as follows:
> ***
> **[Q1] Regarding quantitative results:**
>
> Thanks for your insightful comments. Image smoothing reduces image contents over the input image. Therefore, there are no general agreement objective metrics to directly measure image quality for smooth images. We can use no-reference image quality metrics to evaluate the performance of smoothing downstream tasks, which indirectly illustrates the performance of smoothing methods.
>
> **We utilize five no-reference image quality objective evaluation metrics: BRISQUE [1], PIQE [2], SSEQ [3], ILNIQUE [4], and CEIQ [5] to compare performance on image detail enhancement and HDR tone mapping tasks**. The smaller values of BRISQUE and PIQE denote higher-quality images. While the larger values of SEQ, ILNIQUE, and CEIQ indicate higher-quality images. The quantitative results corresponding to Figure 6 are in the following Table:
> | Methods | BRISQUE $\downarrow$ | PIQE $\downarrow$ | SSEQ $\uparrow$ | ILNIQUE $\uparrow$ |
> |:------:|:----:|:-----:|:--------:|:------:|
> | L0   | 10.6974  | 35.6176     | 24.7566| 117.81  |
> | ILS | 23.5255    | 37.0118  | 17.7950 | 122.43  |
> | DRTV | 8.5623      | 33.5745 | 25.1130  | 124.46  |
> | RoG | 14.9437   | 34.6882  | 24.5738| 124.32 |
> | muGIF | 20.6328  | 36.6683  | 25.1297|  121.44  |
> | dRTV   | 15.4301  | 25.1630  | 9.9543 | 124.49  |
> | Ours  | **7.6912**   | **24.9118**    | **26.7940**         | **128.84**          |
> ***
> We present the quantitative numerical results corresponding to Figure 14 in the following Table:
> | Methods | BRISQUE$\downarrow$ | PIQE $\downarrow$ | SSEQ $\uparrow$ | ILNIQUE$\uparrow$ |
> |:----:|:-----:|:-----:|:------:|:-----:|
> | RGF     | 28.8617 | 30.1807 | 23.7323 | 117.56 |
> | ResNet  | 22.3801| 36.4881| 26.4587 | 120.12 |
> | muGIF   | 16.0524 | 30.2385 | 19.8289 | 123.25 |
> | VDCNN   | 28.9499  | 35.4522| 26.1938| 123.85|
> | Ours    | **9.6503** | **26.8804**| **31.5996** | **126.78**|
> ***
> We present the quantitative numerical results corresponding to Figure 15 in the following Table:
> | Methods | BRISQUE$\downarrow$ | PIQE$\downarrow$ | SSEQ $\uparrow$ | ILNIQUE$\uparrow$ | CEIQ$\uparrow$ |
> |:---:|:-----:|:----:|:------:|:-----:|:----:|
> | L0      | 24.8086  | 28.6853| 12.3523 | 129.59| 2.6947|
> | dRTV    | 23.7972 | 26.9161| 16.8464| 129.16| 2.6867|
> | RGF     | 24.3680 | 27.2629 | 15.6265 | 126.34 | 2.7728 |
> | Ghosh   | 23.7344  | 38.8357 | 15.3685| 125.49| 2.0085|
> | RoG     | 28.1463 | 28.8937 | 12.1442| 135.53| 2.8496|
> | muGIF   | 22.8831| 32.4163 | 13.2606 | 132.94| 2.6515 |
> | RTV     | 20.7134  | 30.0749  | 16.1376  | 133.17| 2.6804 |
> | Ours    | **14.8449** | **24.8736**| **17.9714** | **135.69** | **2.8867** |
> ***
> These results demonstrate that our model achieves significant superiority over other methods in downstream tasks of smoothed images.
> ***
> **References:**
>
> [1] Mittal, Anish, Anush Krishna Moorthy, and Alan Conrad Bovik. "No-reference image quality assessment in the spatial domain." IEEE Transactions on image processing 21.12 (2012): 4695-4708.
>
> [2] Venkatanath, Narasimhan, et al. "Blind image quality evaluation using perception based features." 2015 twenty first national conference on communications (NCC). IEEE, 2015.
>
> [3] Liu, Lixiong, et al. "No-reference image quality assessment based on spatial and spectral entropies." Signal processing: Image communication 29.8 (2014): 856-863.
>
> [4] Zhang, Lin, Lei Zhang, and Alan C. Bovik. "A feature-enriched completely blind image quality evaluator." IEEE Transactions on Image Processing 24.8 (2015): 2579-2591.
>
> [5] Yan, Jia, Jie Li, and Xin Fu. "No-reference quality assessment of contrast-distorted images using contrast enhancement." arXiv preprint arXiv:1904.08879 (2019).

---

> > ### Author Response · Authors · 2024-11-22
> >
> > **[Q2] Regarding the exploration of specific visual tasks:**
> >
> > We are grateful for your thoughtful suggestions. Image smoothing is one of the fundamental image processing tasks and plays a crucial role in various visual tasks. In our manuscript, we have introduced three specific visual tasks, including detail enhancement, HDR tone mapping, and comparison artifact removal.
> >
> > 1. For image **detail enhancement tasks**, edges and main structures preserved after image smoothing are vital for detail enhancement that can reduce halo artifacts and gradient reversals. We have presented detailed illustrations in line 417 of the Experiment Section.
> >
> > 2. For **compression artifact removal tasks**, the goal is to remove compression blocks along edges via smoothing. The edges and main structures preserved are essential for keeping complete information of the input image. The related description can be found in the Section compression artifact removal in line 431 of our manuscript.
> >
> > 3.  For **HDR tone mapping tasks**,  the more edges and main structures preserved, the HDR tone mapping image can produce fewer staircase artifacts along edges and halo artifacts. We have presented detailed explanations in line 1003 of our manuscript.
> >
> > It is worth noting that the proposed model is **not suitable for super-resolution tasks**, but this is a **great potential research topic, prompting that we can extend our model to super-resolution tasks** by designing specific priors in future work.
> > ***
> > Thank you for helping us improve the clarity and completeness of our paper.

---

### Official Review · Reviewer_kXjP · 2024-11-03

**Soundness:** 3
**Presentation:** 3
**Contribution:** 3
**Rating:** 6
**Confidence:** 5

**Summary:**

The author introduces a mutually guided edge-aware smoothing model based on relative wavelet domain representation. Their proposed RWDR serves as a novel measure for effectively differentiating between textures and structures.

**Strengths:**

1. The solution of the proposed model is supported by a complete theoretical guarantee, which is a strong point.
2. Extensive experiments prove that the proposed method outperforms existing algorithms in mitigating gradient reversals,
 staircase artifacts, and halos and achieves a superior performance in balancing smoothing strength and edge preservation.

**Weaknesses:**

1. Though the authors support their claims by extensive qualitative results, but they should also provide the quantitative results to validate their points in the main paper or at least in the supplementary. For instance, the authors can include PSNR (Peak Signal-to-Noise Ratio), SSIM (Structural Similarity Index), or MSE (Mean Squared Error) on standard synthetic benchmark datasets and LPIPS, MUSIQ, NIQE, MANIQA for real-world tasks. This would allow for a more objective comparison with existing method.
2. The method section needs to be refined, as mentioned in Fig1 (that the detail enhancement image is boosted by four detail layers), this statement is not explained in the method section, how the four detail layers are being generated , is it from the wavelet decomposition?
3. The paper has lacks ablation study. The authors have given mathematical proofs of choosing the particular operations like RWDR and the edge-aware scale map, they should also try to prove the effectiveness of each proposed component  on the overall model.
4. The authors should also try to report the results on some real-world applications in Super-Resolution (RealSR, DrealSR,RealLR200), denoising (SIDD, DND) that would further prove the use of the proposed model, if the time permits and should also check on synthetic SR datasets like Manga109, Urban100, and BSD68 (for denoising).

**Questions:**

Please check the weakness section.

---

> ### Author Response · Authors · 2024-11-22
>
> We sincerely thank Reviewer kXjP for their careful review and insightful questions. We respectfully address each concern below:
> ***
> **[Q1] Regarding the quantitative results:**
>
> Thanks for your thoughtful suggestions. It is worth noting that there are no general agreement objective metrics to directly measure image quality for smooth images. However, we can use no-reference image quality metrics to evaluate the performance of smoothing downstream tasks, which indirectly illustrates the performance of smoothing methods. We have utilized PSNR and SSIM to evaluate the smoothing performance in the artifact removal task, as shown in Table 1. The MSE metric has been adopted to illustrate smoothing performance in Figure 8. To provide more quantitative results to validate our points,  **we utilize five no-reference image quality objective evaluation metrics: BRISQUE [1], PIQE [2], SSEQ [3], ILNIQUE [4], and CEIQ [5] to compare performance on image detail enhancement and HDR tone mapping tasks**. The smaller values of BRISQUE and PIQE denote higher-quality images. While the larger values of SEQ, ILNIQUE, and CEIQ indicate higher-quality images. The quantitative results corresponding to Figure 6 are in the following Table:
> | Methods | BRISQUE $\downarrow$ | PIQE $\downarrow$ | SSEQ $\uparrow$ | ILNIQUE $\uparrow$ |
> |:------:|:----:|:-----:|:--------:|:------:|
> | L0   | 10.6974  | 35.6176     | 24.7566| 117.81  |
> | ILS | 23.5255    | 37.0118  | 17.7950 | 122.43  |
> | DRTV | 8.5623      | 33.5745 | 25.1130  | 124.46  |
> | RoG | 14.9437   | 34.6882  | 24.5738| 124.32 |
> | muGIF | 20.6328  | 36.6683  | 25.1297|  121.44  |
> | dRTV   | 15.4301  | 25.1630  | 9.9543 | 124.49  |
> | Ours  | **7.6912**   | **24.9118**    | **26.7940**         | **128.84**          |
> ***
> We present the quantitative numerical results corresponding to Figure 14 in the following Table:
> | Methods | BRISQUE$\downarrow$ | PIQE $\downarrow$ | SSEQ $\uparrow$ | ILNIQUE$\uparrow$ |
> |:----:|:-----:|:-----:|:------:|:-----:|
> | RGF     | 28.8617 | 30.1807 | 23.7323 | 117.56 |
> | ResNet  | 22.3801| 36.4881| 26.4587 | 120.12 |
> | muGIF   | 16.0524 | 30.2385 | 19.8289 | 123.25 |
> | VDCNN   | 28.9499  | 35.4522| 26.1938| 123.85|
> | Ours    | **9.6503** | **26.8804**| **31.5996** | **126.78**|
> ***
> We present the quantitative numerical results corresponding to Figure 15 in the following Table:
> | Methods | BRISQUE$\downarrow$ | PIQE$\downarrow$ | SSEQ $\uparrow$ | ILNIQUE$\uparrow$ | CEIQ$\uparrow$ |
> |:---:|:-----:|:----:|:------:|:-----:|:----:|
> | L0      | 24.8086  | 28.6853| 12.3523 | 129.59| 2.6947|
> | dRTV    | 23.7972 | 26.9161| 16.8464| 129.16| 2.6867|
> | RGF     | 24.3680 | 27.2629 | 15.6265 | 126.34 | 2.7728 |
> | Ghosh   | 23.7344  | 38.8357 | 15.3685| 125.49| 2.0085|
> | RoG     | 28.1463 | 28.8937 | 12.1442| 135.53| 2.8496|
> | muGIF   | 22.8831| 32.4163 | 13.2606 | 132.94| 2.6515 |
> | RTV     | 20.7134  | 30.0749  | 16.1376  | 133.17| 2.6804 |
> | Ours    | **14.8449** | **24.8736**| **17.9714** | **135.69** | **2.8867** |
> ***
> These results demonstrate that our model achieves significant superiority over other methods in downstream tasks of smoothed images.
> ***
> **References:**
>
> [1] Mittal, Anish, Anush Krishna Moorthy, and Alan Conrad Bovik. "No-reference image quality assessment in the spatial domain." IEEE Transactions on image processing 21.12 (2012): 4695-4708.
>
> [2] Venkatanath, Narasimhan, et al. "Blind image quality evaluation using perception based features." 2015 twenty first national conference on communications (NCC). IEEE, 2015.
>
> [3] Liu, Lixiong, et al. "No-reference image quality assessment based on spatial and spectral entropies." Signal processing: Image communication 29.8 (2014): 856-863.
>
> [4] Zhang, Lin, Lei Zhang, and Alan C. Bovik. "A feature-enriched completely blind image quality evaluator." IEEE Transactions on Image Processing 24.8 (2015): 2579-2591.
>
> [5] Yan, Jia, Jie Li, and Xin Fu. "No-reference quality assessment of contrast-distorted images using contrast enhancement." arXiv preprint arXiv:1904.08879 (2019).
> ***
> **[Q2] Regarding the four detail layers in Figure 1:**
>
> We are grateful for your insightful comments. Figure 1 shows the visual effects of the image detail enhancement. It aims to enhance high-frequency regions by incorporating a detail layer into the input image. The core of this technology involves **extracting the high-frequency detail layer by subtracting the smoothed image from the original input**. In other words, **the four details are made by repeating four times the extracted high-frequency detail layer, which is not from the wavelet decomposition**. We presented the detailed process and visual results in the image detail enhancement part of our manuscript. We have added the above-mentioned description in Figure 1.

---

> > ### Author Response · Authors · 2024-11-22
> >
> > **[Q3] Regarding the ablation study:**
> >
> > We greatly appreciate your suggestions. The detailed ablation experiments of the proposed RWDR and the edge-aware scale map have been added in the Experiment section of our manuscript. **To assess the capability of RWDR in distinguishing between textures and structures, we conduct an ablation study on RWDR as shown in Figure 8. ** The model deployed without RWDR has mistreated texture as structure, leading to removing texture uncleanly. In contrast, **to assess the capability of the edge-aware scale map in edge preservation, we conduct an ablation study on the edge-aware scale map as shown in Figure 9.** The model deployed without the edge-aware scale map has smoothed textures cleanly while making main structures and edges lost and blurred.
> > ***
> > **[Q4] Regarding the real-world applications:**
> >
> > We sincerely thank your questions. We have presented **three real-world smoothing applications** in our manuscript, including **image detail enhancement, clip-art compression artifact removal, and HDR tone mapping**. The three real-world applications further prove the significant superiority of the proposed model over other methods.
> >
> > Image smoothing is to remove textures and perturbations that are larger than noise in size. **It is worth noting that the proposed model is not suitable for super-resolution and denoising tasks.** Additionally, the rebuttal time is not enough for us to extend our model to super-resolution and denoising tasks. However, these are **great potential research topics, prompting that we can further extend our model to super-resolution and denoising tasks in future work.**
> > ***
> > We sincerely appreciate your contributions in providing insightful advice to help us improve the quality of our manuscript.

---

### Official Review · Reviewer_bdtH · 2024-11-03

**Soundness:** 3
**Presentation:** 4
**Contribution:** 3
**Rating:** 6
**Confidence:** 3

**Summary:**

The main contribution of this work is the introduction of RWDR that effectively distinguishes textures from primary structures and preserves weaker edges. Additionally, the paper proposes an innovative edge-aware scale map method that dynamically adjusts scale based on the image structure, resulting in clearer distinctions between structure and texture. Experimental results demonstrate that the proposed approach provides superior edge-preserving smoothing compared to existing methods.

**Strengths:**

1. This paper introduces relative wavelet domain representation into bilateral filtering, which is reasonable and novel.

2. The method achieves superior visual results compared to previous studies.

3. The paper includes comprehensive theoretical derivations, technical descriptions, and runtime analysis of the algorithm.

**Weaknesses:**

1. The paper provides extensive visual results, but I’m curious how different algorithms are objectively evaluated based on visual quality. The authors should consider comparing performance on downstream tasks with objective metrics. A user study could also statistically confirm the advantages of the proposed method.

2. As a new method, it likely performs well in certain scenarios. However, I am more interested in its robustness and stability. In other words, can the authors provide a lower bound for the algorithm's performance? In which scenarios might it fail? Additionally, how sensitive is the algorithm to parameter changes?

**Questions:**

Could the authors provide an online demo to allow users to test the method easily? While it’s not essential for acceptance, it would add value for potential users.

---

> ### Author Response · Authors · 2024-11-22
> **Official Comment by Authors**
>
> We appreciate Reviewer bdtH for their detailed feedback and thoughtful questions. We respectfully address each concern below:
> ***
> **[Q1] Regarding the objective metrics:**
>
> Thanks for your suggestions, we reviewed a large amount of relevant literature and found that no objective metrics can be directly used to measure the quality of smoothed images. For the downstream tasks, **we utilize five no-reference image quality objective evaluation metrics: BRISQUE [1], PIQE [2], SSEQ [3], ILNIQUE [4], and CEIQ [5] to compare performance on image detail enhancement and HDR tone mapping tasks**. The smaller values of BRISQUE and PIQE denote higher-quality images. While the larger values of SEQ, ILNIQUE, and CEIQ indicate higher-quality images. The quantitative results corresponding to Figure 6 are in the following Table:
> | Methods | BRISQUE $\downarrow$ | PIQE $\downarrow$ | SSEQ $\uparrow$ | ILNIQUE $\uparrow$ |
> |:--------------:|:----------------------------:|:--------------------------:|:------------------------:|:--------------------------:|
> | L0             | 10.6974                      | 35.6176                    | 24.7566                  | 117.81                     |
> | ILS            | 23.5255                      | 37.0118                    | 17.7950                  | 122.43                     |
> | DRTV           | 8.5623                       | 33.5745                    | 25.1130                  | 124.46                     |
> | RoG            | 14.9437                      | 34.6882                    | 24.5738                  | 124.32                     |
> | muGIF          | 20.6328                      | 36.6683                    | 25.1297                  | 121.44                     |
> | dRTV           | 15.4301                      | 25.1630                    | 9.9543                   | 124.49                     |
> | Ours           | **7.6912**              | **24.9118**          | **26.7940**         | **128.84**          |
>
> ***
> We present the quantitative numerical results corresponding to Figure 14 in the following Table:
> | Methods | BRISQUE$\downarrow$ | PIQE $\downarrow$ | SSEQ $\uparrow$ | ILNIQUE$\uparrow$ |
> |:-------:|:-------------------:|:-----------------:|:---------------:|:-----------------:|
> | RGF     | 28.8617             | 30.1807           | 23.7323         | 117.56            |
> | ResNet  | 22.3801             | 36.4881           | 26.4587         | 120.12            |
> | muGIF   | 16.0524             | 30.2385           | 19.8289         | 123.25            |
> | VDCNN   | 28.9499             | 35.4522           | 26.1938         | 123.85            |
> | Ours    | **9.6503**              | **26.8804**           | **31.5996**         | **126.78**      |
> ***
> We present the quantitative numerical results corresponding to Figure 15 in the following Table:
> | Methods | BRISQUE$\downarrow$ | PIQE$\downarrow$ | SSEQ $\uparrow$ | ILNIQUE$\uparrow$ | CEIQ$\uparrow$ |
> |:-------:|:-------------------:|:----------------:|:---------------:|:-----------------:|:--------------:|
> | L0      | 24.8086             | 28.6853          | 12.3523         | 129.59            | 2.6947         |
> | dRTV    | 23.7972             | 26.9161          | 16.8464         | 129.16            | 2.6867         |
> | RGF     | 24.3680             | 27.2629          | 15.6265         | 126.34            | 2.7728         |
> | Ghosh   | 23.7344             | 38.8357          | 15.3685         | 125.49            | 2.0085         |
> | RoG     | 28.1463             | 28.8937          | 12.1442         | 135.53            | 2.8496         |
> | muGIF   | 22.8831             | 32.4163          | 13.2606         | 132.94            | 2.6515         |
> | RTV     | 20.7134             | 30.0749          | 16.1376         | 133.17            | 2.6804         |
> | Ours    | **14.8449**             | **24.8736**          | **17.9714**         | **135.69**            | **2.8867**         |
> ***
> These results demonstrate that our model achieves significant superiority over other methods in downstream tasks of smoothed images.
> ***
> **References:**
>
> [1] Mittal, Anish, Anush Krishna Moorthy, and Alan Conrad Bovik. "No-reference image quality assessment in the spatial domain." IEEE Transactions on image processing 21.12 (2012): 4695-4708.
>
> [2] Venkatanath, Narasimhan, et al. "Blind image quality evaluation using perception based features." 2015 twenty first national conference on communications (NCC). IEEE, 2015.
>
> [3] Liu, Lixiong, et al. "No-reference image quality assessment based on spatial and spectral entropies." Signal processing: Image communication 29.8 (2014): 856-863.
>
> [4] Zhang, Lin, Lei Zhang, and Alan C. Bovik. "A feature-enriched completely blind image quality evaluator." IEEE Transactions on Image Processing 24.8 (2015): 2579-2591.
>
> [5] Yan, Jia, Jie Li, and Xin Fu. "No-reference quality assessment of contrast-distorted images using contrast enhancement." arXiv preprint arXiv:1904.08879 (2019).

---

> > ### Author Response · Authors · 2024-11-22
> >
> > **[Q2] Regarding the user study:**
> >
> > We greatly appreciate your suggestions. We carefully consider your suggestion about using a user study to statistically confirm the advantages of our model. **A user study needs to adopt enough amount of samples and sort out statistical questionnaires of users, which takes a lot of time. However, the rebuttal time is not enough to do that, we will add the user study to the extension journal version of our work if our manuscript is accepted.**
> > ***
> > **[Q3] Regarding the fail scenario:**
> >
> > Thanks for your detailed feedback and thoughtful questions. **Since there are no objective metrics to directly measure smoothing performance, we can not provide a quantitative lower bound for our algorithm's performance.** However, we have presented the limitations of our model in the Conclusion part. The proposed RWDR model also faces the challenge of addressing long-range texture dependencies like other state-of-the-art methods. **In other words, the RWDR model fails in the specific scenario with large irregular multiscale textures.**
> > ***
> > **[Q4] Regarding the sensitivity analysis:**
> >
> > Thanks for your comments. We have induced detailed settings of all parameters in our model, as presented in our appendix **B.1**.
> >
> > 1.  **For these parameters with fixed settings**, the proposed algorithm is not sensitive to their changes. Therefore, we provide the **recommended values for most scenarios**.
> >
> > 2. For the rest parameters of our model, the performance of the proposed model is sensitive to their changes. **Values of these unfixed parameters should be fine-tuned according to the texture complexity.** We also give the **recommended range** of these parameter settings in our manuscript.
> > ***
> > **[Q5] Regarding the online demo:**
> >
> > We greatly appreciate your suggestions about providing an online demo to allow users to test the method easily. We have **uploaded our source code in the Supplementary Material**. At the current stage, users can test the method via source code. The **web demo is building**, we believe it will be released to the public soon.
> > ***
> > We have clarified these points in the rebuttal version. Thank you again for your contributions in helping us improve our paper.

---

> > > ### Comment · Reviewer_bdtH · 2024-11-26
> > >
> > > Thanks for the rebuttal. I suggest adding an analysis of failure cases in the final version. My score remains unchanged.

---

> > > > ### Author Response · Authors · 2024-11-26
> > > >
> > > > Thanks for your suggestion. We have added the analysis of failure cases in the final version **(as shown in Figure 17 and Section B.8 of the supplementary document).**

---

### Author Response · Authors · 2024-11-23
**Response Summary**

We sincerely appreciate all reviewers' thorough and constructive comments. We are pleased that the reviewers recognized our method as **reasonable and novel (Reviewer bdtH)**, **complete theoretical guarantee and superior performance (Reviewer kXjP)**, and our **fluent and academic writing (Reviewer M1ZN)**.

Our main responses are as following five folds:

1. Provided comprehensive **objective qualitative results** in the tables below;

2. Clarified the **generation mechanism of the detail layer** in Figure 1;

3. Conducted **detailed ablation studies** in Figure 8 and Figure 9;

4. Clarified our method's **parameters sensitive analysis and recommended settings**;

5. Provided the **specific visual task analysis** and the importance of our approach in real-world applications.

Notably, all these improvements are incorporated into our revised manuscript while maintaining clarity and technical depth.
***
We are grateful for helping us improve our manuscript’s quality and completeness.

---

### Meta-Review · Area_Chair_sJdH · 2024-12-18

**Metareview:**

This paper proposes an effective Relative Wavelet Domain Representation (RWDR) for edge-preserving image smoothing. Experimental results show that the proposed method preserve main edges well.

The major concerns of reviewers including adding more experimental results (e.g., subjective evaluations) and limitation analysis.

In the rebuttal, the authors solve the concerns of reviewers. Based on the recommendations of reviewers, the paper can be accepted.

**Additional Comments On Reviewer Discussion:**

The major concerns of reviewers including adding more experimental results (e.g., subjective evaluations) and limitation analysis. During the discussion stage, the reviewers are satisfied with the response of the authors.

---

### Decision · Program_Chairs · 2025-01-22

Accept (Poster)